# Transcriptomic analysis of intestine following administration of a transglutaminase 2 inhibitor to prevent gluten-induced intestinal damage in celiac disease

Transglutaminase 2 (TG2) plays a pivotal role in the pathogenesis of celiac disease (CeD) by deamidating dietary gluten peptides, which facilitates antigenic presentation and a strong anti-gluten T cell response. Here, we elucidate the molecular mechanisms underlying the efficacy of the TG2 inhibitor ZED1227 by performing transcriptional analysis of duodenal biopsies from individuals with CeD on a long-term gluten-free diet before and after a 6-week gluten challenge combined with 100 mg per day ZED1227 or placebo. At the transcriptome level, orally administered ZED1227 effectively prevented gluten-induced intestinal damage and inflammation, providing molecular-level evidence that TG2 inhibition is an effective strategy for treating CeD. ZED1227 treatment preserved transcriptome signatures associated with mucosal morphology, inflammation, cell differentiation and nutrient absorption to the level of the gluten-free diet group. Nearly half of the gluten-induced gene expression changes in CeD were associated with the epithelial interferon-γ response. Moreover, data suggest that deamidated gluten-induced adaptive immunity is a sufficient step to set the stage for CeD pathogenesis. Our results, with the limited sample size, also suggest that individuals with CeD might benefit from an *HLA-DQ2*/*HLA-DQ8* stratification based on gene doses to maximally eliminate the interferon-γ-induced mucosal damage triggered by gluten.

Gluten-containing cereals are essential foods worldwide. However, in up to 2% of individuals[1], the ingestion of dietary gluten results in an abnormal immune response in the small intestine and the development of celiac disease (CeD). Predisposing genotypes (human leukocyte antigen (HLA), for example, *HLA-DQ2* and *HLA-DQ8*) are necessary but not sufficient for the manifestation of CeD. Diarrhea, weight loss and malnutrition are classical bowel-related symptoms and signs of CeD, but anemia, osteoporosis and other autoimmune diseases, such as type 1 diabetes, are also frequent manifestations[2-4].

Currently, a gluten-free diet (GFD) is the only accepted treatment option for individuals with CeD. However, the life-long strict and restrictive GFD is onerous and difficult to follow, and inadvertent gluten ingestion is common[5,6], resulting in ongoing symptoms in nearly 50% of treated individuals[7,8]. Keeping the GFD also has a big impact on quality of life[9]. Inadvertent gluten ingestion often leads to ongoing duodenal mucosal injury, with inflammation and morphological changes[10]. Thus, even individuals on a GFD frequently have nutrient imbalances and deficiencies[11,12]. We have shown that despite having normal duodenal

e-mail: keijo.viiri@tuni.fi

histomorphology, individuals with CeD on a GFD differ from individuals without CeD on the molecular level and display insufficient expression of micronutrient transporter genes[13]. Thus, adjunctive pharmacological therapy, together with a strict GFD, is needed to efficiently treat CeD.

The CeD autoantigen transglutaminase 2 (TG2) is expressed in the intestine, where it deamidates certain neutral glutamine residues to negatively charged glutamic acid residues in immunogenic gluten peptides[14–16]. These modified gluten peptides are more efficiently presented by HLA-DQ2 or HLA-DQ8 molecules on mucosal antigen-presenting cells, which leads to the activation and expansion of gluten-specific CD4[+] type 1 helper T cells and the secretion of proinflammatory cytokines[17,18]. Eventually, this process leads to villus atrophy, crypt hyperplasia and the production of TG2 IgA.

TG2, being crucial for CeD pathogenesis, is a pertinent target for therapy, and this approach was recently tested in a phase 2, randomized, double-blind, placebo-controlled, dose-finding gluten challenge trial using the oral TG2 inhibitor ZED1227 (ref. 19). In this phase 2 trial, ZED1227 attenuated gluten-induced duodenal mucosal injury, both morphological deterioration and inflammation, and improved symptoms and quality of life scores in individuals with CeD[19]. Here, we report the results of the molecular histomorphometry assessment of ZED1227 efficacy along with intestinal mucosal transcriptomic analysis. Moreover, as the gene dose of *HLA-DQ2* was shown to influence the severity of CeD[20,21], we analyzed the efficacy parameters of ZED1227 relative to the *HLA-DQ2* gene dose.

## Results

### ZED1227 prevents gluten-induced transcriptomic changes

Duodenal biopsies were collected from 58 individuals with CeD before (GFD) and after a 6-week gluten challenge combined with treatment with 100 mg of the TG2 inhibitor ZED1227 per day (postgluten challenge drug (PGCd); *n* = 34) or placebo (PGC placebo (PGCp); *n* = 24). RNA extracted from the 116 biopsy samples was subjected to transcriptomic next-generation sequencing (NGS) analysis.

Principal component analysis (PCA) performed on all samples using DESeq2-transformed counts of all genes showed a moderate level of separation between groups (GFD drug (GFDd), GFD placebo (GFDp), PGCd and PGCp; Fig. 1a). The PGCp group was clearly discernible, whereas the GFDd, GFDp and PGCd groups tended to cluster closer together. There was a clear cosegregation of transcriptomic profiles and mucosal morphology. Thus, a ratio of villus height to crypt depth (VH:CrD) of <1.2 separated from VH:CrD of ≥1.2 and overlapped with PGCp in the PCA (Fig. 1a). A comparison of the PGCp versus GFDp groups detected 95 differentially expressed genes (DEGs; Fig. 1b,c). Strikingly, only one DEG was detected when the GFDd group was compared to the PGCd group, whereas the comparison of the PGCp and PGCd groups indicated 180 DEGs (Fig. 1b,c and Supplementary Data 1).

Because treating participants with ZED1227 eliminated the gluten-induced gene expression changes entirely, it can be assumed that the majority of the DEGs in the PGCp versus GFDp and PGCp versus PGCd comparisons were shared. Indeed, 56 of 95 (59%) DEGs after the gluten challenge were also differentially expressed, according to the comparison of the PGCp and PGCd groups (Fig. 1d). This analysis suggests that a significant number of genes were 'uniquely' differentially expressed after gluten challenge (39 of 95) and between the PGCd and PGCp groups (124 of 180; Fig. 1d). Closer inspection of both 'uniquely expressed' DEGs revealed that they were not uniquely differentially expressed in PGCd but, to an extent, were equivalent to those expressed in the GFD group, although this was not sufficiently statistically significant (for example, due to inadequate log (fold change) (FC) or expression level), relative to the PGCp group (Supplementary Fig. 1). When all detected gene $\log_2$ (FC) values from the PGCp versus GFDp comparison were compared to those from the PGCp versus PGCd comparison, there was a positive correlation, suggesting a similar pattern of gene expression changes in both groups (Fig. 1e). Accordingly, a Pearson's pairwise

correlation heat map analysis with the 220 selected genes showed that the GFDd, GFDp and PGCd groups had similar features, whereas the PGCp group significantly differed from all groups (Fig. 1f). Similar to the results in Fig. 1a, ranking samples according to VH:CrD ratio made it evident that individuals with the most severe mucosal damage, that is, the lowest VH:CrD ratio, had a very different transcriptomic profile (Fig. 1f).

### ZED1227 sustains molecularly assessed intestinal functions

An analysis of the expression data of the 95 DEGs individually after the gluten challenge in the placebo group showed that the expression levels correlated with the VH:CrD ratio (Fig. 2a). Reactome enrichment analysis showed that genes involved in the cellular response to interferon (IFN) signaling, both type 1 (IFNα/IFNβ) and type 2 (IFNγ), were upregulated and overrepresented in the gluten-induced gene expression profile (Fig. 2b, left, and Supplementary Data 2). Transcription motif analyses also indicated that genes harboring motifs for transcription factors transducing IFN signaling (for example, STAT1, RELA and IRF1) were significantly present (Supplementary Fig. 2). Notably, a reactome enrichment comparison of the DEGs in the PGCp versus PGCd groups revealed that the type 2 IFNγ signaling term was no longer statistically significant (Fig. 2b, right, and Supplementary Data 2). Similarly, the Gene Ontology term analyses showed that IFN-mediated inflammatory signaling was enriched in the gluten-induced gene expression profile (Fig. 2c and Supplementary Data 2).

As gluten challenge impairs enterocyte differentiation and absorptive functions and increases inflammation, we analyzed how ZED1227 protects these cellular processes. Gene sets were formed based on human duodenal single-cell RNA-sequencing data[22]. Gene set *z* (GSZ) scores[23] were calculated for each sample. Samples in the PGCd group demonstrated the same GSZ score levels in the categories of transit-amplifying cells, mature enterocytes, immune cells and duodenal transporters as samples in the pooled GFDd and GFDp groups (GFDd + p) group (Fig. 2d). Importantly, the PGCp group was consistently significantly different from the PGCd group, indicating that ZED1277 efficiently sustained intestinal functions to a level similar to that observed in individuals in the GFDd + p group. Bulk RNA-sequencing deconvolution that used duodenal single-cell RNA-sequencing data as a reference revealed similar patterns in cell proportion distributions, like a decrease in enterocyte numbers accompanied with a small increase in stem and Paneth cell numbers in the PGCp group (Supplementary Fig. 3a,b). At the same time, markers for cytotoxic intraepithelial lymphocytes (IELs) seemed to not be altered (except HLA-E) by placebo and drug treatment (Supplementary Fig. 3c), probably because of underrepresentation of these cell types in biopsy samples.

### ZED1227 can halt the IFNγ response

Reactome and Gene Ontology enrichment analyses (Fig. 2b,c) indicated that IFN signaling was one of the most significantly affected pathways in the gluten challenge. Interestingly, a 100-mg dose of ZED1227 for 6 weeks seemed somewhat insufficient in decreasing the IFNγ response, at least according to the Reactome enrichment analysis (Fig. 2b). We decided to set up an intestinal epithelium-specific IFNγ response gene set to assess how well ZED1227 could inhibit inflammation using an epithelial-specific IFNγ response as a gauge. Human intestinal organoids composed of pure intestinal epithelium were treated with IFNγ, and a DEG set was analyzed against the DEGs induced by gluten challenge. We found that nearly half (43 of 95) of the gluten-induced gene expression changes in CeD were associated with the epithelial response to IFNγ (Fig. 3a and Supplementary Data 3). The GSZ scores calculated based on these 43 genes showed that, on average, ZED1227 inhibited the epithelial IFNγ response, as participants in the PGCd group had significantly lower GSZ scores than participants in the PGCp group (Fig. 3b). However, when the GSZ scores of the PGCd and GFDd + p

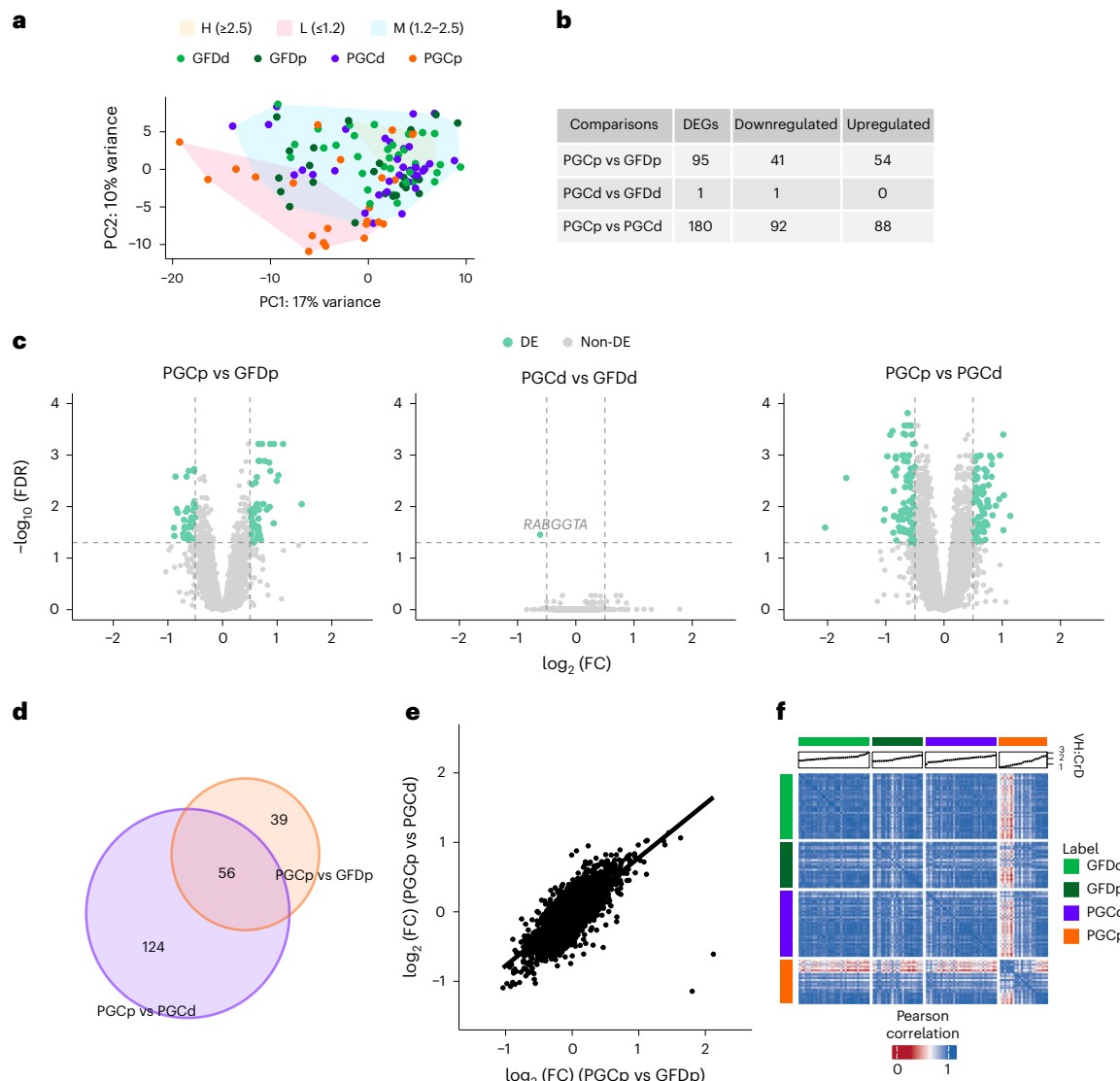

**Fig. 1 | ZED1227 can effectively avert gluten challenge-induced transcriptomic changes in the intestine. a**, PCA plot using DESeq2-transformed counts for all samples ($n = 115$). Green, dark green, violet and orange circles correspond to GFDd ($n = 34$), GFDp ($n = 24$), PGCd ($n = 34$), and PGCp ($n = 23$) samples, respectively. Yellow, blue and red shaded areas depict samples with a high (H; >2.5), medium (M; 1.2–2.5) and low (L; <1.2) range of VH:CrD, respectively. **b**, Table showing the number of DEGs ($\log_2$ (FC) ≥ | 0.5 | and false discovery rate (FDR) ≤ 0.05) in the indicated comparisons. **c**, Volcano plot representations comparing DEGs as indicated. The green dots indicate DEGs (FDR ≤ 0.05) above the threshold ($\log_2$ (FC) of ≥0.5 and ≤−0.5). The dashed horizontal line represents the FDR threshold of 0.05, and the vertical dashed lines represent the $\log_2$ (FC) thresholds (≥| 0.5 |). **d**, Venn diagram illustrating the number of DEGs that are shared in the PGCp versus PGCd and PGCp versus GFDp comparisons. **e**, Correlation profile of all detected gene ($n = 10,063$) $\log_2$ (FC) values between PGCp and GFDp and PGCp and PGCd comparisons. **f**, Pearson's pairwise correlation heat map analyses of 220 DEGs visualizing the cross-correlations of the transcriptomic profiles of the samples (total $n = 115$; GFDd $n = 34$; GFDp $n = 24$; PGCd $n = 34$; PGCp $n = 23$). Samples are organized in the ranking order of increasing VH:CrD ratio (indicated in the scatter charts above the heat map).

groups were compared, there was a slight but statistically significant difference. This suggests that either there was a residual IFNγ response in all/many participants in the PGCd group or ZED1227 was not able to inhibit the IFNγ response completely in some individuals. When GSZ scores were calculated for each sample, it was evident that some individuals (4 of 34 participants in the PGCd group) still had an active epithelial IFNγ response even after the high-dose (100-mg) ZED1227 treatment for 6 weeks (Fig. 3c).

IFNγ has been shown to induce TG2 activity in intestinal epithelial cancer cells, and this has been suggested to contribute to CeD pathogenesis[24]. Similarly, participants in the placebo group after the gluten challenge and concomitant IFNγ response had significantly higher expression of *TGM2*, whereas in participants treated with ZED1227, *TGM2* was expressed at a level similar to that observed in participants

in the GFDd group (Fig. 3d). Overproduced interleukin-21 (IL-21) in CeD is known to sustain IFNγ production[25], and we also detected an induction in the IL-21 signaling pathway in participants in the PGCp group (Supplementary Fig. 4a,b), but this was not statistically significant. We also found that the expression of *TGM2* was positively correlated ($R = 0.65$) with the epithelial IFNγ response (Fig. 3f). Direct causality was further proven by treating human intestinal duodenal organoids with IFNγ, which resulted in a significant induction of *TGM2* mRNA expression (Fig. 3e) that could not be inhibited with ZED1227 treatment (Supplementary Fig. 4c). IFNγ treatment induced TG2 activity in Caco-2 cells, which was inhibited by ZED1227 to the level observed following mock treatment (Supplementary Fig. 4d). These observations could be explained by ZED1227 cell impermeability[26] and its binding mainly to enterocyte luminal surfaces[27].

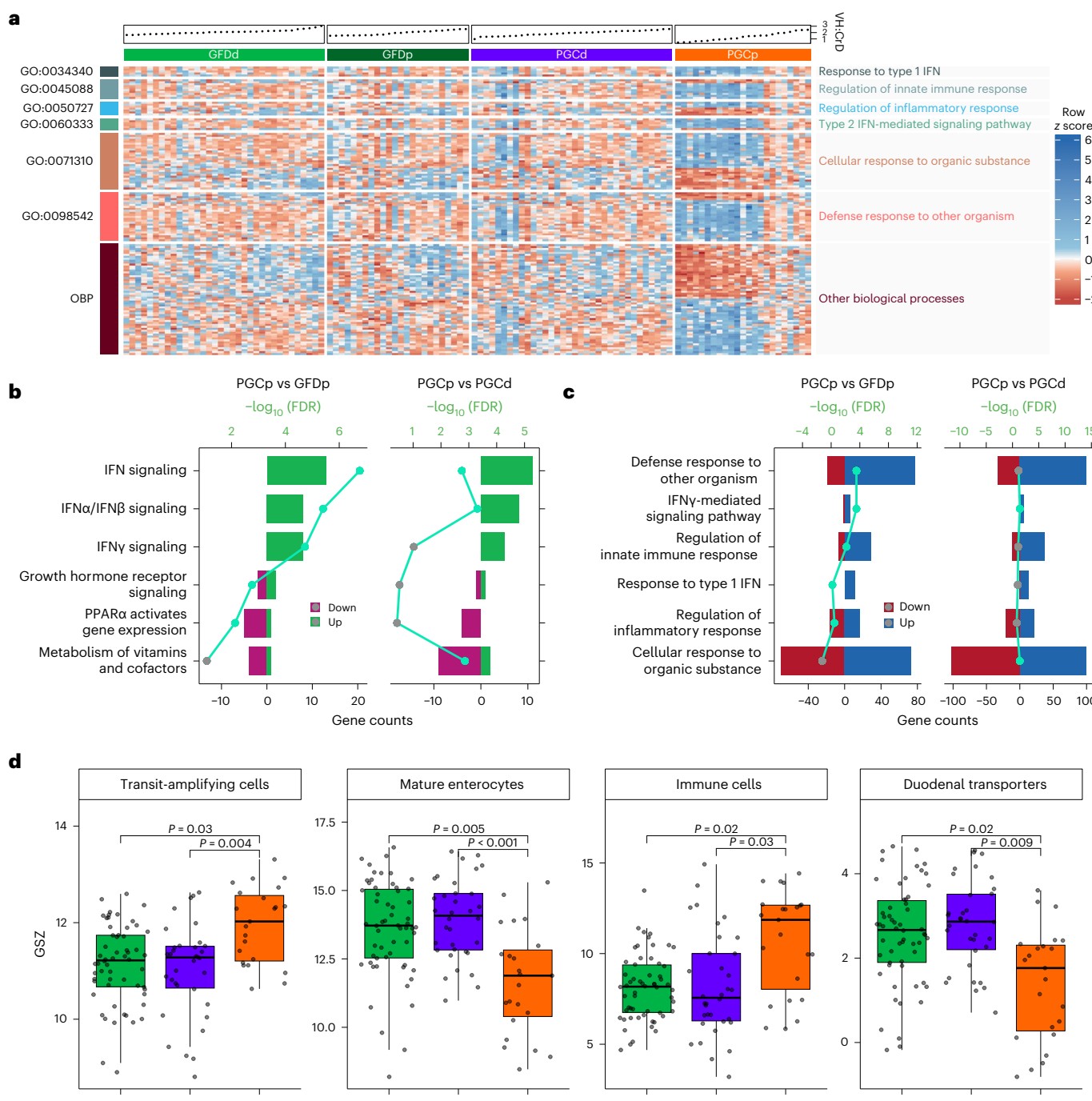

**Fig. 2 | ZED1227 preserves intestinal functions in individuals with CeD while on gluten challenge. a**, Heat map of the 95 DEGs in the PGCp versus GFDp comparison. Samples are ordered by increasing VH:CrD ratio, as depicted in the scatter charts above the heat map (GFDd $n = 34$; GFDp $n = 24$; PGCd $n = 34$; PGCp $n = 23$). Genes are clustered according to Gene Ontology annotation. The $z$-score of normalized expression is plotted; OBP, other biological processes. **b**, Bar plot showing enriched Reactome terms of DEGs in the PGCp group relative to the GFDp and PGCd groups. Enriched terms were determined by overrepresentation analysis. $P$ values were calculated by hypergeometric distribution (one-tailed test) and adjusted for multiple testing using the Benjamini–Hochberg method. Reactome terms with an FDR of <0.05 ($-\log_{10}$ (FDR) > 1.3) were considered enriched. Green and gray dots denote significant and nonsignificant FDRs, respectively. **c**, Bar plots showing Gene Ontology biological process overrepresentation of DEGs in the PGCp group relative to the GFDp and PGCd groups. A Fisher's exact overrepresentation test (one tailed) was used to find enriched categories. The obtained $P$ values were adjusted for multiple testing

using the Benjamini–Hochberg method. Gene Ontology terms with an FDR of <0.05 ($-\log_{10}$ (FDR) > 1.3) were considered enriched. Green and gray dots denote significant and nonsignificant FDRs, respectively. **d**, GSZ score analyses were performed for categories including transit-amplifying cells, mature enterocytes, immune cells and duodenal transporters and are presented as box plots, with center lines representing the median, the box boundaries representing the interquartile range and the whiskers representing the minimum and maximum values. Values from individual participants are shown (GFDd + p $n = 58$; PGCd $n = 34$; PGCp $n = 23$). GSZ scores were compared among groups using asymptotic $P$ value estimation, with statistical significance defined as a $P$ value of <0.05 (transit-amplifying cells: GFDd + p–PGCd $P = 0.3$, PGCp–GFDd + p $P = 0.03$, PGCd–PGCp $P = 0.004$; mature enterocytes: GFDd + p–PGCd $P = 0.3$, PGCp–GFDd + p $P = 0.005$, PGCd–PGCp $P = 5.35 \times 10^{-4}$; immune cells: GFDd + p–PGCd $P = 0.73$, PGCp–GFDd + p $P = 0.02$, PGCd–PGCp $P = 0.03$; duodenal transporters: GFDd + p–PGCd $P = 0.53$, PGCp–GFDd + p $P = 0.02$, PGCd–PGCp $P = 0.009$).

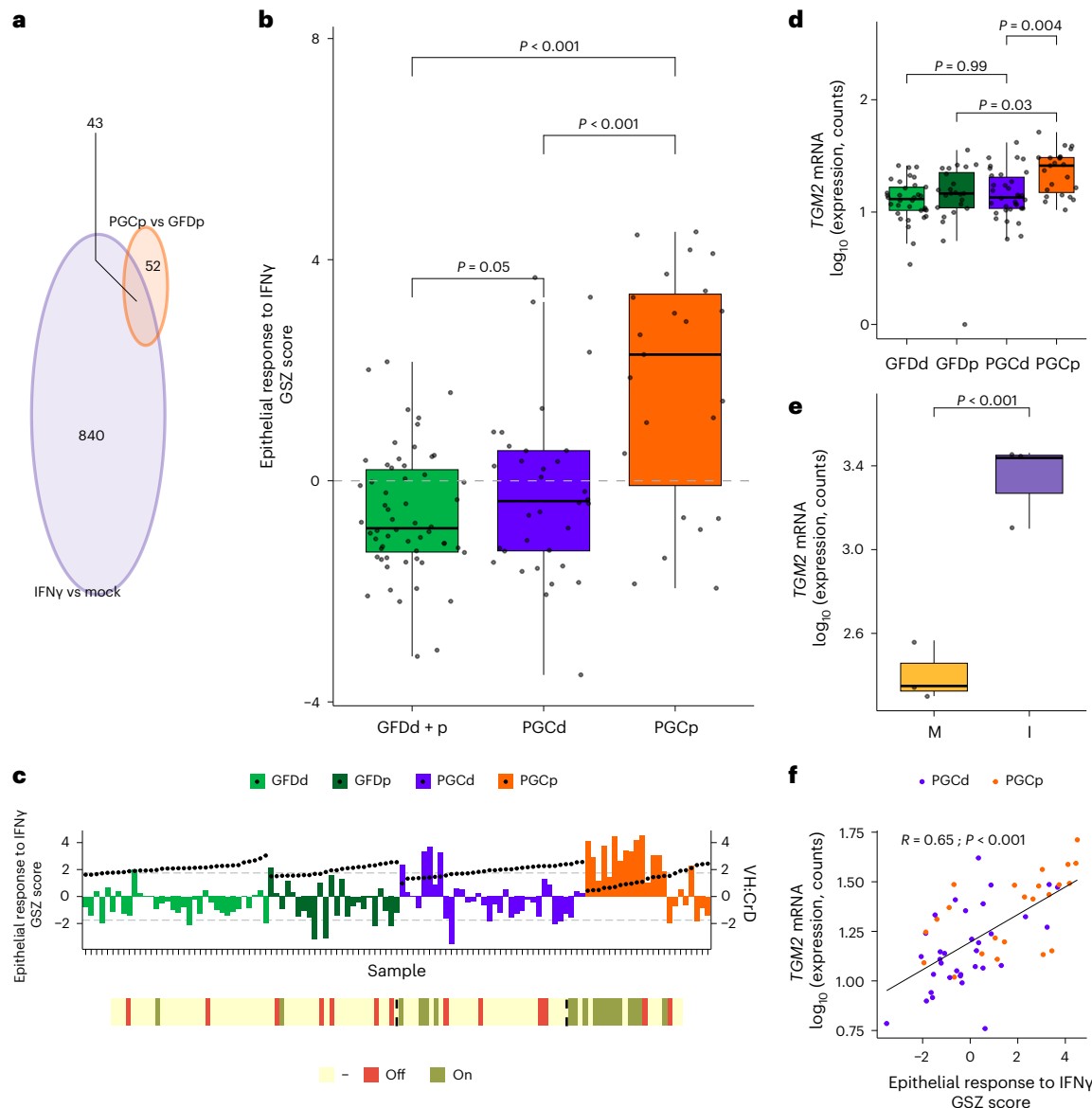

**Fig. 3 | Comparing transcriptomic signatures from CeD biopsies and IFNγ-treated human duodenal organoids. a**, Venn diagram of all DEGs in human duodenal organoids ($n = 3$) after a 24-h treatment with 100 U ml$^{-1}$ IFNγ (violet sphere) and PGCp versus GFD (orange sphere) comparisons. **b**, GSZ score analyses for the epithelial IFNγ-related gene set (GFDd + p $n = 58$; PGCd $n = 34$; PGCp $n = 23$). The box plot center lines represent the median, the box boundaries represent interquartile range, and the whisker length represents the minimum and maximum range. Values from individual participants are shown. GSZ scores were compared among groups using asymptotic $P$ value estimation, with statistical significance defined as a $P$ value of <0.05 (GFDd + p–PGCd $P = 0.05$, $P$GCp–GFDd + p $P = 6.07 \times 10^{-6}$, PGCd–PGCp $P = 1.24 \times 10^{-4}$). **c**, Bar plot of epithelial IFNγ-related GSZ scores calculated for each sample. The dashed lines represent the threshold, outside of which the gene set was considered to be 'on' or 'off'. The yellow bar below illustrates the samples in which the epithelial IFNγ-related GSZ scores were on and off (GFDd $n = 34$; GFDp $n = 24$; PGCd $n = 34$; PGCp $n = 23$). **d**, Expression of *TGM2* mRNA in the GFDd, GFDp, PGCd and PGCp groups. The box plot center lines represent the median, the box boundaries

represent interquartile range, and the whisker length represents the minimum and maximum range. Values from individual participants are shown. Likelihood ratio test (LRT) $P$ values were calculated using DESeq2, with $P$ values representing adjusted values for multiple testing using the Benjamini–Hochberg method (FDR; GFDd $n = 34$; GFDp $n = 24$; PGCd $n = 34$; PGCp $n = 23$). **e**, Expression of *TGM2* mRNA in human duodenal organoids ($n = 3$) treated with 100 U ml$^{-1}$ IFNγ (I) or mock treated (M) for 24 h. The box plot center lines represent the median, the box boundaries represent interquartile range, and the whisker length represents minimum and maximum range. Values from individual participants are shown. LRT $P$ values were calculated using DESeq2, with $P$ values representing adjusted values for multiple testing using the Benjamini–Hochberg method (FDR; $P = 9.48 \times 10^{-17}$). **f**, Correlation plot for *TGM2* mRNA expression and epithelial IFNγ-related GSZ scores. Each dot represents an individual participant with CeD after gluten challenge. Pearson correlation coefficient values ($R$) are presented, and the $P$ value ($P$) was calculated based on the $t$-distribution under the null hypothesis of no correlation using a two-tailed test; $P = 5.57 \times 10^{-8}$.

## ZED1227 prevents activation of gluten-induced immunological pathways

As gluten challenge caused a significant IFNγ response and concomitant upregulation of *TGM2* expression and activity, we analyzed gluten challenge-induced immunological pathway alterations and how ZED1227 can inhibit them. Peroxisome proliferator-activated

receptor-γ (PPARγ) has been shown to transrepress inflammatory responses[28,29]. PPARγ is downregulated in celiac mucosa[30], and this has been shown to be mediated by TG2 and gliadin[31]. We also found that *PPARG* gene expression (Fig. 4a) and the corresponding signaling pathway (Fig. 4b) are significantly less active after gluten challenge in the PGCp group than in the GFD and PGCd groups. We also observed

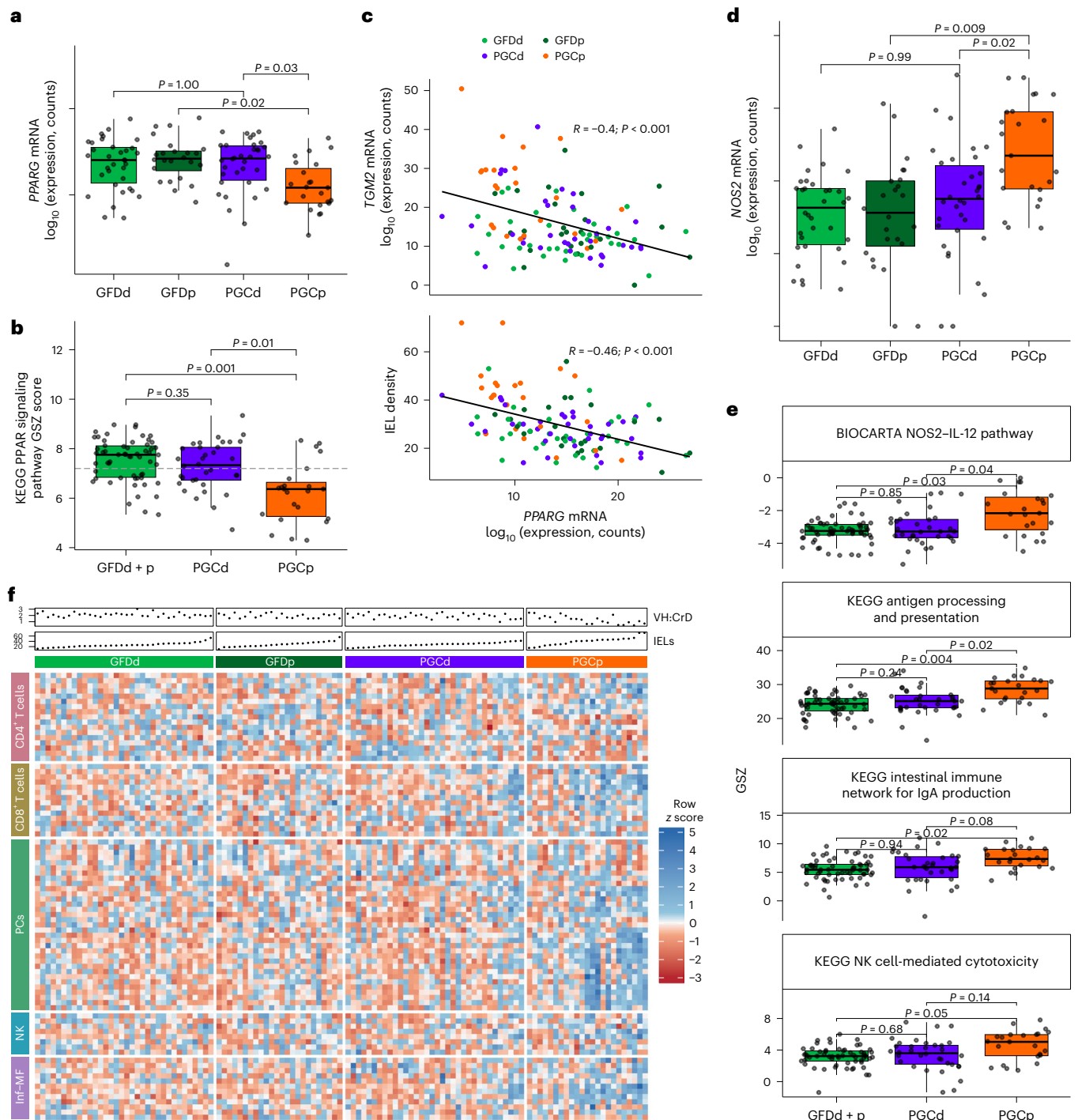

**Fig. 4 | Effects of ZED1227 treatment on immunological pathways.**
**a**, Expression of *PPARG* mRNA in the GFDd, GFDp, PGCd and PGCp groups. LRT *P* values were calculated using DESeq2, with *P* values representing adjusted values for multiple testing using the Benjamini–Hochberg method (FDR; GFDd *n* = 34; GFDp *n* = 24; PGCd *n* = 34; PGCp *n* = 23). **b**, GSZ score analyses for the PPAR signaling pathway from the KEGG database gene set. GSZ scores were compared among groups using asymptotic *P* value estimation, with statistical significance defined as a *P* value of <0.05 (GFDd + p *n* = 58; PGCd *n* = 34; PGCp *n* = 23). **c**, Correlation plots for *TGM2* mRNA expression (top) and IEL density (number of CD3$^+$ cells per 100 enterocytes; bottom) against *PPARG* mRNA expression. Each dot represents an individual participant with CeD after gluten challenge. The Pearson correlation coefficient (*R*) is presented, and the *P* value (*P*) was calculated based on the *t*-distribution under the null hypothesis of no correlation using a two-tailed test (*TGM2* mRNA expression versus *PPARG* mRNA expression, $P = 1.14 \times 10^{-5}$; IEL density versus *PPARG* mRNA expression, $P = 2.95 \times 10^{-7}$).

**d**, Expression of *NOS2* mRNA in the GFDd, GFDp, PGCd and PGCp groups. LRT *P* values were calculated using DESeq2, with *P* values representing adjusted values for multiple testing using the Benjamini–Hochberg method (FDR; GFDd *n* = 34; GFDp *n* = 24; PGCd *n* = 34; PGCp *n* = 23). **e**, GSZ score analyses for selected KEGG, BIOCARTA and Reactome database gene sets. GSZ scores were compared among groups using asymptotic *P* value estimation, with statistical significance defined as a *P* value of <0.05 (GFDd + p *n* = 58; PGCd *n* = 34; PGCp *n* = 23). The box plot center lines represent the median, the box boundaries represent interquartile range, and the whisker length represents minimum and maximum range. Values from individual participants are shown. **f**, Heat map for selected CeD-specific immune cell marker genes detected in Atlasy et al.[58]. Samples are ordered by increasing IEL density, as depicted in the scatter charts above the heat map (GFDd *n* = 34; GFDp *n* = 24; PGCd *n* = 34; PGCp *n* = 23). The *z* scores of normalized expression are plotted; PC, plasma cells; Inf-MF, inflammatory macrophages.

**Table 1 | Distribution of *HLA-DQ* genotypes in individuals with CeD in drug and placebo groups**

| | Drug (*n*=34) | | Placebo (*n*=24) | |
|---|---|---|---|---|
| | *n* | % | *n* | % |
| G1 | 6 | 17.6 | 2 | 8.3 |
| *DQ2.5* homozygous | 5 | 14.7 | 1 | 4.2 |
| *DQ2.5/DQ2.3* | 1 | 2.9 | 0 | 0.0 |
| *DQ2.5*/one copy of *DQB1*02* | 0 | 0.0 | 1 | 4.2 |
| G2 | 14 | 41.2 | 6 | 25.0 |
| *DQ2.2* homozygous | 4 | 11.8 | 2 | 8.3 |
| *DQ8* homozygous | 6 | 17.6 | 1 | 4.2 |
| *DQ8/DQ2* | 2 | 5.9 | 2 | 8.3 |
| *DQ2.5/DQ2.2* | 2 | 5.9 | 1 | 4.2 |
| G3 | 14 | 41.2 | 15 | 62.5 |
| *DQ2* half heterodimer | 8 | 23.5 | 6 | 25.0 |
| *DQ2.2* heterozygous | 1 | 2.9 | 4 | 16.7 |
| *DQ2.5* heterozygous | 4 | 11.8 | 3 | 12.5 |
| *DQ8* heterozygous | 1 | 2.9 | 1 | 4.2 |
| *DQ2.5*/one copy of *DQA1*05* | 0 | 0.0 | 1 | 4.2 |
| Not identified | 0 | 0.0 | 1 | 4.2 |

a negative correlation between the expression of *TGM2* and *PPARG* and the expression of *PPARG* and IEL count (Fig. 4c). This suggests that the mucosal inflammatory response, kept in check by PPARγ, is lifted during the gluten challenge in CeD, and this can be prevented with ZED1227 treatment.

PPARγ inhibits the expression of proinflammatory cytokines, and it also silences inducible nitric oxide (NO) synthase (iNOS/NOS2)[32]. NOS2 is induced in the mucosa of individuals with active CeD mainly in macrophages and enterocytes[33–35], leading to a systemic increase of NO in the plasma[36].

NO is needed for the responsiveness of natural killer (NK) cells to the NK cell-activating factor IL-12, which stimulates cytotoxicity and IFNγ release[37]. Our data show that ZED1227 can inhibit gluten challenge-induced *NOS2* upregulation (Fig. 4d), resulting in overrepresentation of gene sets involved in the NO–IL-12 and NK cell-mediated cytotoxicity (Fig. 4e) pathways. Also, pathways to antigen presentation and IgA production are normalized following ZED1227 treatment (Fig. 4e). Analysis of the expression of immunological cell gene markers showed that ZED1227 inhibits the infiltration of cell types (especially CD8+ T cells, plasma cells, NK cells and macrophages) involved in the aforementioned inflammatory responses (Fig. 4f).

## The effect of *HLA-DQ* genetic background on treatment outcomes

The fact that some participants treated with ZED1227 in the PGCd group still showed a significant epithelial IFNγ response (Fig. 3c), as a sign of active residual CeD pathophysiology prompted us to study factors behind the incomplete response to treatment. To this end, we performed high-resolution genotyping for HLA class II *DQ* alleles using the arcasHLA tool[38] from aligned sequences obtained from genome-wide 3′ RNA-sequencing data. Five participants had too low coverage either at the *HLA-DQB1* or *HLA-DQA1* locus, according to RNA sequencing; thus, their allele typing was performed from blood samples collected at the on study inclusion. One participant from the placebo group, however, failed during identification. This participant is marked as 'not identified' in Table 1 and was excluded from subsequent analyses.

It is known that *HLA-DQ2* gene dose correlates with the strength of the gluten-specific T cell response[20]; thus, all obtained genotypes

were divided into groups by their potential effectiveness in binding and presenting gliadins to T cells[39,40]. We were able to divide participants into three groups according to their *HLA-DQ* genotypes, with G1 being the high-gluten-response group and G3 being the low-gluten-response group (Table 1). However, one should note that the group sizes are relatively small.

When examining the changes in mean VH:CrD ratio within genotype groups over time (Fig. 5a), it is evident that the groups exhibit different trajectories of change. Notably, the slope of the G1 group appears to deviate the most from the parallel pattern among the groups for both drug and placebo treatments.

The impact of treatment on VH:CrD ratio within different time points (GFD and PGC) across *HLA-DQ* genetic background groups (G1, G2 and G3) was assessed by fitting repeated-measures analysis of variance (ANOVA). In the placebo group, the interaction term between time point and *HLA-DQ* genetic groups was statistically significant ($P = 0.003$; Table 2 and Methods), indicating that *HLA-DQ* genetic background has an impact on changes in VH:CrD ratio over the course of gluten challenge (Methods). For the drug group, however, the interaction term was not significant ($P = 0.06$; Table 2 and Methods), suggesting that the drug appears to be effective in reducing the impact of gluten across all genotype groups. However, pairwise comparisons (Table 2 and Methods) performed for the drug group showed that the impact of *HLA-DQ* genetic background is statistically significant for the G1 group ($P = 0.05$) and not significant for the G2 ($P = 0.07$) and G3 groups ($P = 0.39$).

Given the notable drop in the VH:CrD ratio after ZED1227 treatment in the high-gluten-response genotype group (G1), we analyzed the efficacy of treatments in each genotype group. A two-way analysis of covariance (ANCOVA) was performed to examine the effects of treatment and *HLA-DQ* genetic background on VH:CrD ratio at PGC. After adjustment for the VH:CrD ratio at GFD, there was no statistically significant interaction between treatment and the *HLA-DQ* genotype group on the histomorphometry parameters (Methods), and pairwise multiple comparisons show significant differences between the PGC VH:CrD means in all genotype groups between participants receiving drug or placebo (Fig. 5b). This suggests that, despite a substantial decrease in VH:CrD ratio after gluten challenge in the G1 group for participants treated with drug, the VH:CrD ratio was still higher in the drug group than in the placebo group, irrespective of the genotype.

The estimated difference in the VH:CrD ratio for participants treated with drug belonging to the G3 genotype versus the G1 genotype was −0.52 (95% CI of −0.86 to −0.19) with an adjusted *P* value of 0.01, as assessed by fitting a one-way ANCOVA model. Other estimated differences (G3–G2 and G2–G1) were not significant but showed the tendency of group G2 having the intermediate position between G1 and G3, when judging by VH:CrD ratio (Fig. 5c). Interestingly, the G1 high-risk genotype specifically affected VH and not CrD (Extended Data Fig. 2a,b).

The CeD pathophysiological epithelial IFNγ response was studied with a two-way ANCOVA statistical analysis, and pairwise comparisons showed that participants in the PGCd and G1 genotype groups still had an active IFNγ response and did not statistically differ from the placebo group (Fig. 5d). In fact, in the bar plot presenting four participants in the PGCd group with an IFNγ response in Fig. 3c, three of these participants had the high-gluten-response genotype homozygous *HLA-DQ2.5* and one had homozygous *HLA-DQ8* associated with an intermediate response to gluten.

The inclination of the G1 group to be highly responsive to gluten and less reactive to ZED1227 was also observed at individual gene expression levels. Reduced expression of enterocyte marker genes (*APOB*, *APOA1* and *TM4SF4*) and increased expression of proliferation markers (*AGR2*, *MKI67* and *CENPF*) were observed in participants with G1 genotypes in both the ZED1227- and placebo-treated groups (Fig. 5e). Inflammation-related genes (*STAT1*, *GBP1* and *TGM2*) showed lower expression in PGCd samples with G2 and G3 genotypes, suggesting

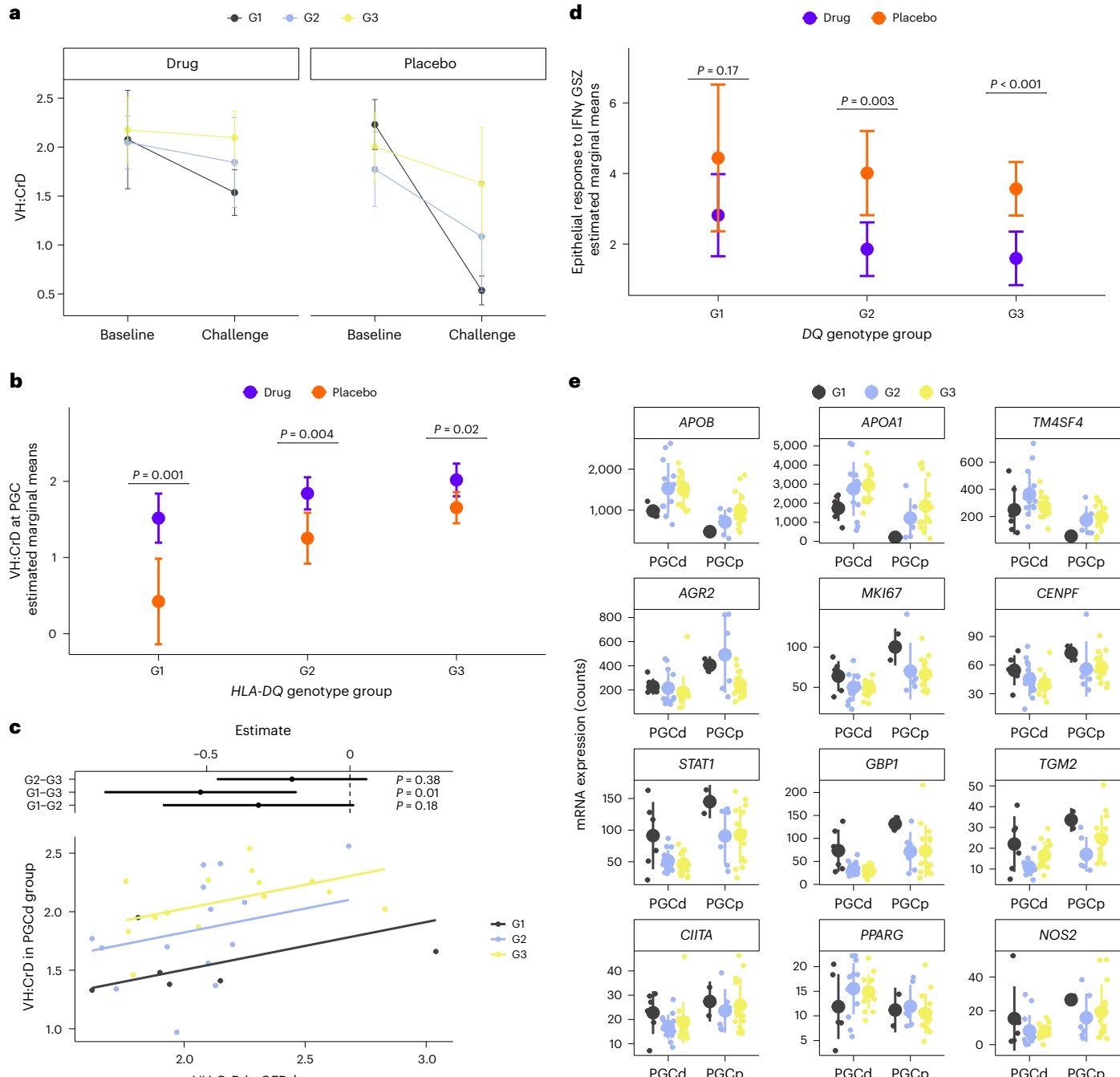

**Fig. 5 | Effects of *HLA-DQ* genetic background on VH:CrD ratio and gene expression. a**, The VH:CrD ratio remains higher in the drug group than in the placebo group, regardless of the genotype. Participants (*n* = 57) were divided into two groups according to the treatment received (drug or placebo). The VH:CrD ratio at PGC is shown as mean ± s.d. **b**, A two-way ANCOVA was performed with the VH:CrD ratio at PGC as a dependent variable, the VH:CrD ratio at GFD as a covariate and treatment (drug *n* = 34 and placebo *n* = 23) and *HLA-DQ* genotype group (G1, G2 and G3) as independent variables (ANCOVA, $F_{2,50}$ = 2.2, *P* = 0.12). Post hoc pairwise multiple comparisons were performed between the drug and placebo groups among *HLA-DQ* genotype groups. The VH:CrD ratio at PGC is shown as estimated marginal means ± 95% confidence interval (95% CI; drug G1 *n* = 6; drug G2 *n* = 14; drug G3 *n* = 14; placebo G1 *n* = 2; placebo G2 *n* = 6; placebo G3 *n* = 15). **c**, The G1 genotype group showed weaker recovery after ZED1227 treatment as assessed by VH:CrD ratio. Participants (*n* = 34) belonging to the drug group were selected for one-way ANCOVA. The VH:CrD ratio at PGCd was used as a dependent variable, and the VH:CrD ratio

at GFD was used as a covariate; *HLA-DQ* genotype group (G1, G2 and G3) served as independent variables (ANCOVA, $F_{2,30}$ = 5.11, *P* = 0.012). Post hoc pairwise multiple comparisons were performed between *HLA-DQ* genotype groups, with *P* values adjusted by Bonferroni correction. Results are shown as estimate ± 95% CI. **d**, A two-way ANCOVA plot examining the effects of treatment and *HLA-DQ* genetic background on PGC epithelial response to IFNγ GSZ score (ANOVA, $F_{2,49}$ = 0.07, *P* = 0.93). The epithelial response to IFNγ GSZ score at PGC is shown as estimated marginal means ± 95% CI (drug G3 versus placebo G3 *P* = $5.50 \times 10^{-4}$; drug G1 *n* = 6; drug G2 *n* = 14; drug G3 *n* = 14; placebo G1 *n* = 2; placebo G2 *n* = 6; placebo G3 *n* = 15). **e**, Expression of enterocyte- (*APOB*, *APOA1* and *TMSF4*), proliferation- (*AGR2*, *MKI67* and *CENPF*) and inflammation-related (*STAT1*, *GBP1*, *TGM2*, *CIITA*, *PPARG* and *NOS2*) marker genes. Expression is shown as counts grouped by *HLA-DQ* genotype group (G1, G2 and G3) and are presented as mean (spheres) and s.d. (vertical lines; PGCd G1 *n* = 6; PGCd G2 *n* = 14; PGCd G3 *n* = 14; PGCp G1 *n* = 2; PGCp G2 *n* = 6; PGCp G3 *n* = 15).

**Table 2 | Summary of changes in VH:CrD values within study time points according to *HLA-DQ2*/*HLA-DQ8* genotype groups**

| Group | *n* | GFD mean±s.d. | PGC mean±s.d. | Change in ratio from GFD (95% CI) | P value |
|---|---|---|---|---|---|
| **Drug** | | | | | |
| G1 | 6 | 2.08±0.5 | 1.54±0.23 | −0.54 (−1.07 to −0.01) | 0.05 |
| G2 | 14 | 2.05±0.27 | 1.84±0.46 | −0.21 (−0.43 to 0.02) | 0.07 |
| G3 | 14 | 2.17±0.33 | 2.1±0.27 | −0.08 (−0.27 to 0.11) | 0.39 |
| **Interaction term time point: *HLA-DQ* genetic group** | | | | 0.06* | |
| **Placebo** | | | | | |
| G1 | 2 | 2.23±0.25 | 0.54±0.15 | −1.69 (−2.65 to −0.74) | 0.03 |
| G2 | 6 | 1.77±0.38 | 1.09±0.56 | −0.69 (−1.23 to −0.15) | 0.02 |
| G3 | 15 | 2.00±0.35 | 1.63±0.57 | −0.38 (−0.62 to −0.13) | 0.005 |
| **Interaction term time point: *HLA-DQ* genetic group** | | | | 0.003* | |

Values are shown as mean±s.d. The change from GFD is presented as a least-squares means estimate. *P* values for interactions are marked with an asterisk (*) and were calculated as part of a repeated-measures ANOVA; other *P* values were obtained from pairwise comparisons using two-tailed *t*-tests.

that they were more susceptible to ZED1227 treatment. In accordance with the higher residual CeD-associated epithelial IFNγ response in participants in the PGCd and G1 groups (Figs. 3b,c and 5d), these inflammatory genes were more highly expressed in participants treated with either placebo or drug within the genotype group G1. Furthermore, ZED1227 was less able to prevent gluten challenge-induced attenuation of PPARγ-mediated inhibition of *NOS2* expression, as the expression of these genes was at the same level in G1 genotypes in the PGCd group as in the G2 and G3 genotypes in the PGCp group (Fig. 5e). Also, the HLA class II transcriptional coactivator *CIITA* was more highly expressed in individuals with the G1 genotype in the PGCd group (Fig. 5e). Moreover, the G1 group was identified as more pathognomonic when its GSZ scores for 'transit-amplifying cells', 'mature enterocytes, 'immune cells' and 'duodenal transporters' were assessed (Extended Data Fig. 2c). In addition to IFNγ signaling, molecular pathways for PPAR and lipid signaling seemed to also be affected in the G1 group (Extended Data Fig. 2c).

**Molecular histomorphometric analysis of ZED1227 efficacy**

We previously created a molecular histomorphometric model to assess gluten-dependent morphological deterioration and healing in the duodenum, that is, VH:CrD, in gene transcriptomic terms[13]. This model is based on the expression of four genes (*ATP8B2*, *PLA2R1*, *PDIA3* and *TM4SF4*), which we showed is significantly correlated with the extent of gluten-induced histological damage[13]. Scatter plots and partial regression plots for these genes showed that the relationship between gene expression and VH:CrD ratio was linear, and participants in the PGCp group tended to separate from participants in the GFD and PGCd groups (Fig. 6a). Moreover, a comparison of traditional and molecular histomorphometry in the regression scatter plot revealed a high coefficient of determination ($R^2 = 0.86$), indicating that the previously developed molecular histomorphometric tool was able to reliably estimate VH:CrD ratios in this independent study cohort (Fig. 6b). Finally, box plot comparisons of groups with histomorphometric and molecular histomorphometric values indicated that ZED1227 efficiently inhibited gluten-induced mucosal damage in individuals with CeD (Fig. 6c).

## Discussion

The ability of the TG2 inhibitor ZED1227 (ref. 26) to attenuate gluten-induced mucosal damage was previously reported in a proof-of-concept, randomized, double-blind, placebo-controlled 6-week trial with a daily 3-g gluten challenge[19]. TG2, the celiac autoantigen[14], has a pivotal role in gluten-induced pathogenesis, leading to small intestinal mucosal injury with villus atrophy and crypt hyperplasia, the histological hallmarks of untreated CeD. Here, we sought to assess the efficacy of ZED1227 in preventing gluten-induced mucosal damage at the transcriptomic level. Remarkably, a 100-mg daily dose of ZED1227 inhibited virtually all gluten-induced transcriptomic changes (Fig. 1b,c). Active CeD is accompanied by compromised enterocyte maturation, crypt hyperplasia due to the expansion of transit-amplifying cells[41–43], immune cell infiltration[44,45] and decreased expression of duodenal transporters[13,46,47]. GSZ[23] scores based on published single-cell databases[22] clearly indicated that TG2 inhibition efficiently blocked all aforementioned gluten-induced intestinal manifestations in individuals with CeD (Fig. 2d). Our recently published molecular histomorphometry regression model based on genome-wide transcriptomics analysis[13] was validated in this independent study sample. We showed a significant accordance between this new molecular tool and the traditional, more subjective biopsy-based microscopic histomorphometry reading. Overall, our transcriptomic findings strongly support the results of the clinical trial with ZED1227, which demonstrated that the inhibition of TG2 activity can efficiently and specifically prevent gluten-induced mucosal damage[19]. Our data also corroborate the previous findings that gliadin together with active TG2 induces attenuated PPARγ activity, which, together with a concomitant increase in IFNγ, lead to increased mucosal NO production and inflammation[30,31,33–36]. We show here that by inhibiting the gliadin deamidation activity of TG2, all these pathogenic immunological changes in CeD can be prevented (Fig. 4). In addition, studies have shown that gluten-derived peptides may have innate immune stimulatory properties, outside the realm of adaptive immunity, which can lead to epithelial stress in CeD[48,49]. Our data show, however, that halting the adaptive immunity pathway in CeD pathogenesis is sufficient to prevent gluten-induced mucosal damage, as we did not detect any molecular traces of mucosal damage remaining after ZED1227 treatment.

Gene Ontology and Reactome analyses indicated that gluten challenge most significantly affected genes related to the immune response, especially IFN-mediated defense mechanisms (Fig. 2b,c). This is in agreement with previously published transcriptomic analyses of individuals with active CeD compared to individuals on a GFD or healthy individuals[47,50,51]. Notably, IFNγ secreted by gluten-reactive T cells in the celiac intestine induces TG2 expression and secretion and thus favors the pathogenic autoamplificatory loop of enhanced gluten deamidation by TG2, improved antigenic presentation on HLA-DQ2 or HLA-DQ8 and subsequent gluten-specific T cell activation[24]. The present study confirms the prominent role of IFN signaling in CeD pathogenesis, in line with findings that nearly half of the gluten-induced gene expression changes in duodenal biopsies can be recapitulated in human intestinal epithelial organoids treated with IFNγ (Fig. 3a). We also detected the suggested autoamplificatory loop in our human data, as *TGM2* expression positively correlated with the epithelial IFNγ response (Fig. 3f). Notably, *TGM2* expression was induced by IFNγ in human intestinal organoids ex vivo (Fig. 3e), suggesting mutual amplification between these two key players in CeD pathogenesis. The functional relevance of this amplification loop was indeed confirmed in the clinical study in which the inhibition of TG2 activity by ZED1227 in individuals with CeD significantly inhibited both the (epithelial) IFNγ response (Fig. 3b) and *TGM2* expression (Fig. 3d), resulting in protection from villous atrophy and intraepithelial lymphocytosis (Fig. 2d).

However, even though TG2 inhibition exhibited significant efficacy, according to a comparison of the transcripts of the placebo/gluten challenge and the gluten challenge/ZED1227-treated group,

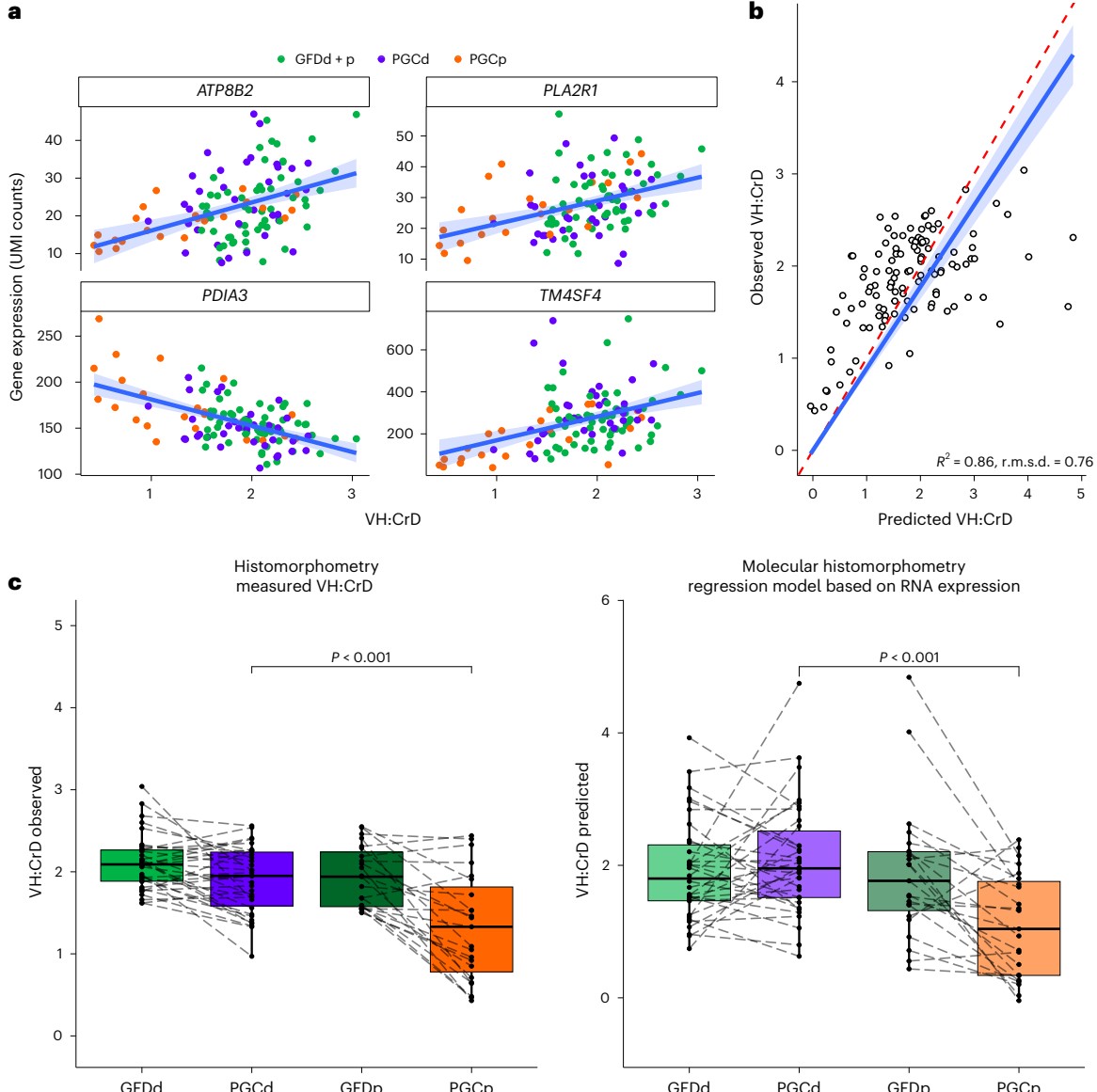

**Fig. 6 | Molecular histomorphometry regression model based on RNA expression. a**, Scatter plots and regression plots for VH:CrD prediction model genes. A linear regression with 95% CI is shown. Each dot represents an individual participant (GFDd + p $n = 58$; PGCd $n = 34$; PGCp $n = 23$); UMI, unique molecular identifiers. **b**, Observed versus predicted regression scatter plot for the model predicting VH:CrD. Each dot represents an individual participant ($n = 115$). A linear regression with 95% CI is shown; $R^2 = 0.86$, $F_{1,114} = 691.6$, $P < 0.001$. The red dashed line represents the ideal regression case, where $x = y$; r.m.s.d., root mean square deviation. **c**, Box plot comparisons of groups with histomorphometry

(measured VH:CrD) values and molecular histomorphometry (regression model based on RNA expression) values. The box plot center lines represent the median, the box boundaries represent interquartile range, and the whisker length represents minimum and maximum range. Values from individual participants are shown. Two-tailed unpaired Student's $t$-tests were used for the PGCd versus PGCp group comparisons (VH:CrD observed: PGCd–PGCp $P = 6.41 \times 10^{-4}$; VH:CrD predicted: PGCd–PGCp $P = 4.04 \times 10^{-5}$; GFDd $n = 34$; GFDp $n = 24$; PGCd $n = 34$; PGCp $n = 23$).

which showed a transcriptome profile similar to that of the GFD groups, we detected heterogeneity regarding the gluten-induced and IFNγ-dependent cascade of pathogenic events among ZED1227-treated and gluten-challenged individuals with CeD. Four of these individuals still had a modestly active IFNγ response, and the majority (three of four) belonged to the *HLA-DQ2.5* homozygous genotype (Fig. 3c). *HLA-DQ2.5* homozygous individuals have a fivefold higher risk of developing CeD than *HLA-DQ2.5* heterozygous individuals[21], which has been linked to the more efficient presentation of deamidated gluten peptides to gluten-specific T cells[20]. Moreover, homozygosity for *HLA-DQ2* predisposes individuals to developing more rapid and severe villous atrophy[52] and is associated with malignant complications, such as refractory CeD type 2 and enteropathy-associated T cell

lymphoma[53,54]. Along this line, we also found that individuals belonging to the high-gluten-response *HLA-DQ* genotype group (G1) were more sensitive to gluten, as their VH:CrD ratios dropped significantly more than individuals belonging to the mid- and low-gluten response groups (G2 and G3) during the gluten challenge, both in the placebo and drug groups (Tables 1 and 2 and Fig. 5a,b). Thus, even after drug treatment, VH:CrD decreased significantly in the G1 versus G2 and G3 genotype groups after the gluten challenge. This was also evident when molecular histomorphometric features were assessed (Extended Data Fig. 2c). We also discovered that PPAR signaling and lipid metabolism, previously reported to be dysregulated in CeD[30], were less controlled in the G1 group (Extended Data Fig. 2c). As IFNγ is known to inhibit PPAR and lipid metabolism[55], it is conceivable that these are consequences of the

overactive IFNγ response in individuals in the G1 group. Nevertheless, duodenal mucosal morphology and, especially, intraepithelial lymphocyte infiltration were significantly healthier in the ZED1227-treated group than in the placebo group, indicating that participants with G1 phenotypes may benefit from a higher dose and/or prolonged treatment with ZED1227. We suggest that the ZED1227 therapy program should include a personalized medicine approach in which *HLA-DQ* stratification is combined with TG2 dose adjustments, which may lead to an optimal treatment response and a more thorough abrogation of IFNγ-induced mucosal damage. According to our transcriptomic analysis of human intestinal organoids, ZED1227 does not appear to induce significant transcriptomic changes in the organoid model (Supplementary Fig. 5), consistent with the clinical safety observed in the phase 2 challenge study[19].

We recognize the limitations of this study. The cohort is relatively modest and characterized by an uneven distribution of *HLA-DQ* genotypes. This resulted in small G1 subgroups within both the drug and placebo cohorts, which may have implications for statistical power and the generalizability of our results and warrants further corroborative studies. Additionally, we only had one dose of the drug available for this study. The transcriptomic analysis was conducted as an optional component of the study, and RNA isolation was not performed for all drug groups. This decision was made to focus our efforts on the drug group that showed the most significant improvement compared to the placebo group, allowing us to investigate potential transcriptomic changes effectively within the study's scope.

In conclusion, the strategy to inhibit TG2 activity as a key upstream effector in gluten-induced immune activation in CeD, which has been proven efficient in the clinical study, was mechanistically buttressed by our transcriptomic analysis of the duodenal biopsies of individuals treated or not treated with ZED1227. Importantly, TG2 inhibition prominently prevented the gluten-induced IFNγ response and further downstream pathways that lead to mucosal inflammation, remodeling and villous atrophy. Our analysis also suggests that, based on *HLA-DQ2.5* genetics, the dose or dose interval of ZED1227 may have to be adjusted for optimal efficacy, but larger sample sizes are required to confirm this assumption. Moreover, CeD-associated gene expression changes were observable, even on a strict GFD[13,56], indicating that complete avoidance of gluten is impossible[5,6]. In fact, a recent meta-analysis found that 15% of foods labeled as gluten free and 28% labeled as naturally gluten free contained more than 20 mg kg$^{-1}$ gluten[57], the cutoff for qualifying as gluten free. Thus, an adjunctive TG2 inhibition-based therapy combined with a GFD would especially benefit highly gluten-sensitive individuals (possibly carrying a homozygous *HLA-DQ* genotype) by providing protection against intestinal damage that can occur even in a low-gluten environment.

## Online content

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

Valeriia Dotsenko [1], Bernhard Tewes[2], Martin Hils [3], Ralf Pasternack[3], Jorma Isola [4,5], Juha Taavela [1,6], Alina Popp[1,7], Jani Sarin[5], Heini Huhtala[8], Pauliina Hiltunen[9], Timo Zimmermann[2], Ralf Mohrbacher [2], Roland Greinwald[2], Knut E. A. Lundin [10,11], Detlef Schuppan [12,13], Markku Mäki [1], Keijo Viiri [1] ✉ & CEC-3 Investigators*

[1]Celiac Disease Research Center, Faculty of Medicine and Health Technology, Tampere University and Tampere University Hospital, Tampere, Finland. [2]Dr. Falk Pharma GmbH, Freiburg, Germany. [3]Zedira GmbH, Darmstadt, Germany. [4]Faculty of Medicine and Health Technology, Tampere University, Tampere, Finland. [5]Jilab Inc, Tampere, Finland. [6]Department of Gastroenterology and Alimentary Tract Surgery, Tampere University Hospital, Tampere, Finland. [7]University of Medicine and Pharmacy 'Carol Davila' and National Institute for Mother and Child Health, Bucharest, Romania. [8]Unit of Health Sciences, Faculty of Social Sciences, Tampere University, Tampere, Finland. [9]Department of Pediatrics, Tampere University Hospital, Tampere, Finland. [10]Norwegian Coeliac Disease Research Centre, Institute of Clinical Medicine, Faculty of Medicine, University of Oslo, Oslo, Norway. [11]Department of Gastroenterology, Oslo University Hospital Rikshospitalet, Oslo, Norway. [12]Institute of Translational Immunology and Celiac Center, Medical Center, Johannes-Gutenberg University, Mainz, Germany. [13]Division of Gastroenterology, Beth Israel Deaconess Medical Center, Harvard Medical School, Boston, MA, USA. *A list of members and their affiliations appears at the end of the paper. ✉e-mail: keijo.viiri@tuni.fi

## CEC-3 Investigators

Karin Kull[14], Jari Koskenpato[15], Mika Scheinin[16], Marja-Leena Lähdeaho[9,17], Michael Schumann[18], Yurdagül Zopf[19], Andreas Stallmach[20], Ansgar W. Lohse[21], Stefano Fusco[22], Jost Langhorst[23,24], Helga Paula Török[25], Valerie Byrnes[26], Juozas Kupcinskas[27], Øistein Hovde[28], Jørgen Jahnsen[29], Luc Biedermann[30] & Jonas Zeitz[30,31]

[14]Department of Gastroenterology, Internal Medicine Clinic, Tartu University Hospital, Tartu, Estonia. [15]Lääkärikeskus Aava Helsinki Kamppi, Helsinki, Finland. [16]Clinical Research Services Turku–CRST Oy, Turku, Finland. [17]Faculty of Medicine and Health Technology, Tampere University and Tampere University Hospital, Tampere, Finland. [18]Department for Gastroenterology, Infectious diseases and Rheumatology, Campus Benjamin Franklin, Charité–University Medicine Berlin, Berlin, Germany. [19]Department of Medicine 1, Hector Center for Nutrition, Exercise, and Sports, Universitätsklinikum Erlangen, Friedrich-Alexander-University Erlangen-Nürnberg, Erlangen, Germany. [20]Department of Internal Medicine IV, Jena University Hospital, Friedrich-Schiller University Jena, Jena, Germany. [21]I. Department of Medicine, University Medical Center Hamburg-Eppendorf, Hamburg, Germany. [22]Department of Gastroenterology, Gastrointestinal Oncology, Hepatology, Infectious Diseases and Geriatrics, University Hospital Tübingen, Tübingen, Germany. [23]Department for Internal and Integrative Medicine, Kliniken Essen-Mitte, Essen, Germany. [24]Department for Internal and Integrative Medicine, Sozialstiftung Bamberg, Medical Faculty, University of Duisburg-Essen, Bamberg, Germany. [25]Department of Medicine II, University Hospital, LMU Munich, Munich, Germany. [26]University College Hospital Galway, Galway, Ireland. [27]Gastroenterology Department and Institute for Digestive Research, Lithuanian University of Health Sciences, Kaunas, Lithuania. [28]Medical Department, Institute of Clinical Medicine, Innlandet Hospital Trust, Gjøvik, Norway. [29]Akershus University Hospital, Lørenskog, Norway. [30]Department of Gastroenterology and Hepatology, University Hospital Zürich, Zurich, Switzerland. [31]Swiss Celiac Center, Center for Gastroenterology, Clinic Hirslanden, Zurich, Switzerland.

## Methods

### Participants and biopsies

PAXgene-fixed and paraffin-embedded biopsies were collected from a multisite, double-blind, randomized, placebo-controlled trial aimed at dose finding and assessing the efficacy and tolerability of a 6-week treatment with ZED1227 capsules versus placebo in individuals with well-controlled CeD undergoing gluten challenge[59]. Full inclusion and exclusion criteria are published[19]. Briefly, participants who had a biopsy-proven CeD diagnosis, were on a self-reported strict GFD for at least 1 year and symptom free, showed normalized duodenal histology compared to the initial diagnostic biopsy finding (morphometrically defined as a mean VH:CrD of 1.5 or higher) and tested negative for serum anti-TG2 on study inclusion were included (GFD group; Extended Data Table 1). These participants then underwent a challenge with a cookie containing 3 g of gluten daily for 6 weeks (PGC group). At least 80% compliance was confirmed[19].

Biopsy sampling was performed twice on study inclusion (denoted here as GFD) and at the final visit (denoted here as PGC; Extended Data Fig. 1). Duodenal forceps biopsies were immersed in PaxFPE (PAXgene fixative) and processed for paraffin block embedding using a standard formalin-free paraffin-infiltration protocol. For morphology, samples were stained with hematoxylin and eosin and measured using our validated morphometry rules separately for morphology (VH, CrD and VH:CrD)[60].

This study used samples from two groups, placebo and the 100-mg ZED1227 group, which represented the highest dose drug group showing the most significant improvement compared to the placebo group. In total, 58 participants (drug group, $n = 34$; placebo group, $n = 24$; total number of biopsies = 116) of the 68 participants who had sufficient biopsy samples at both time points in the original trial[19] were included, as these exploratory (optional) studies required separate written informed consent. Demographic characteristics and duodenal histomorphometry changes in the form of VH:CrD ratio of the participants in the original cohort and in the present study are presented in Supplementary Tables 1 and 2.

### Human organoid cultures

Human duodenal tissues for establishing organoid cultures used in this study were sourced from deidentified surgical specimens ($n = 3$) of the duodenum obtained from participants who had undergone biopsy procedures unrelated to CeD at Tampere University Hospital. The protocol was approved by the Ethics Committee of Tampere University Hospital (ETL code R18082). Intestinal crypts containing stem cells were isolated following 2 mM EDTA dissociation of tissue samples for 30 min at 4 °C (ref. 61). Crypts were washed in PBS, and fractions enriched in crypts were collected. The supernatant was removed, and the crypt epithelial cells were seeded in 50% Matrigel (diluted with basal culture medium). Crypts were passaged and maintained in WELR500 culture medium, as previously described[62]. Organoids were treated with 100 U ml$^{-1}$ IFNγ (Peprotech, 300-02) with or without 50 μM ZED1227 (Zedira) for 24 h and subjected to RNA sequencing to assess any adverse direct side effects to the intestinal epithelium (Supplementary Fig. 5).

### Cell culture and treatments

Caco-2 colonic epithelial cells (ATCC, HTB-37; passage 22–35) were grown as standard monolayers in tissue culture flasks in complete MEM 1 g l$^{-1}$ glucose medium (20% heat-inactivated fetal bovine serum, 1% nonessential amino acids, 1% penicillin–streptomycin, 1% GlutaMAX and 1% sodium pyruvate) at 37 °C in a 5% $CO_2$ atmosphere. Caco-2 cells were treated with 100 U ml$^{-1}$ IFNγ (Peprotech, 300-02) with or without 50 μM ZED1227 (Zedira) or mock treated with DMSO for 24 h. Cells were collected by trypsinization and lysed in lysis buffer (50 mM Tris (pH 8.0), 150 mM NaCl and 1% IGEPAL) supplemented with 0.2 mM DTT and 1× Complete Protease Inhibitor Cocktail (Roche, 11836170001) and used for the transglutaminase activity assay.

### RNA extraction and RNA sequencing

Total RNA was extracted from the PaxFPE-fixed biopsy specimens ($n = 116$)[63] using additional cuttings from the samples on which histomorphometry was previously assessed[19]. For extraction, an RNeasy kit (Qiagen) was used according to the manufacturer's instructions. Library preparation and NGS were performed by the Qiagen NGS Service. A total of 10 ng of purified RNA was converted into NGS cDNA libraries. Library preparation was quality controlled using capillary electrophoresis. Based on the quality of the inserts and the concentration measurements, the libraries were pooled in equimolar ratios and sequenced on a NextSeq (Illumina) sequencing instrument according to the manufacturer's instructions, with 100-bp read length for read 1 and 27-bp read length for read 2. The raw data were demultiplexed, and FASTQ files for each sample were generated using bcl2fastq2 software (Illumina).

RNA from the duodenal organoids was isolated using an RNeasy kit (Qiagen) following the manufacturer's instructions. RNA purity and concentration were measured using a NanoDrop One spectrophotometer (NanoDrop Technologies). Preparation of the RNA library and transcriptome sequencing was conducted by Novogene. mRNA was purified from total RNA using poly(A) selection and subjected to library construction. Sequencing was performed on an Illumina platform, and 150-bp paired-end reads were generated.

### Bioinformatic analyses

Data quality was checked using FastQC. The 3' adapter sequences were trimmed, reads without adapters were kept, and reads with <15 bp were removed. Reads were aligned to the human genome reference consortium human build 38 (GRCh38) using the splice-aware aligner STAR. For all downstream analyses, genes with low expression (read counts that were equal to the number of samples multiplied by 5) were excluded. One sample with low total reads (1.13 million reads) was excluded, leaving 115 samples for subsequent analyses. The mean total reads for all samples were $3.51 \pm 0.07$ million reads. A secondary differential expression analysis involving normalization of unique molecular identifier counts and a subsequent pairwise differential regulation analysis was performed using the DESeq2 package[64]. Pre- and post-treatment samples were compared, and the paired nature of samples was included as a term in the multifactor design formula. The obtained $P$ values were adjusted for multiple testing using the Benjamini–Hochberg method[65]. Genes with an FDR of <0.05 and $|\log_2(FC)|$ of ≥0.5 identified by DESeq2 were assigned as differentially expressed.

Gene Ontology enrichment and Reactome enrichment analyses were performed using topGO[66] and ReactomePA[67] R packages. GSZ scores, as a particular type of overrepresentation analysis, were calculated as previously described[68]. For comparison of groups, mean GSZ score asymptotic $P$ value calculation was applied to our datasets[69]. Gene lists for transit-amplifying cells, mature enterocytes, immune cells and duodenal transporters were retrieved from healthy human duodenal single-cell sequencing analyses published by Busslinger et al.[22] or our DEG analysis from human duodenal organoids treated with IFNγ versus mock-treated organoids. Cell-type proportions for CeD biopsy bulk RNA-sequencing data were estimated with the MuSiC analysis toolkit[70] using single-cell RNA-sequencing data from duodenal adult biopsies[71] as a reference.

Exact HLA genotypes, with a focus on DQ status (*HLA-DQA1* and *HLA-DQB1* alleles), were determined in silico from RNA-sequencing data using the arcasHLA tool[38]. FASTQ files were used as input files. The minimum gene read count required for genotyping was set at 5. Due to low expression, low resolution[72] (Field1, allele group) was taken into consideration in the subsequent statistical analyses.

### Statistical analysis

Statistical tests were conducted as specified in the legends of the respective figures using R version 4.3.0 (R Foundation for Statistical

Computing). A repeated-measures ANOVA was used to assess the impact of treatment on VH:CrD ratio within different time points (GFD and PGC) across *HLA-DQ* genetic background groups (G1, G2 and G3). This analysis comprised 57 participants with identifiable *HLA-DQ* genotypes. Three null hypotheses were proposed: (1) VH:CrD means are equal across time points, (2) VH:CrD means are equal among *HLA-DQ* groups, and (3) there is no interaction between these two factors. As a post hoc analysis, multiple pairwise *t*-tests were used to identify differences between time points for each genotype group. To assess how the impact of the *HLA-DQ* genotype group on the VH:CrD outcome varies with different time points, a one-way ANOVA model was used. To address multiple testing, a Bonferroni correction was applied to *P* values (total tests performed = 2). Statistical significance was determined as $P < 0.05$.

To assess the interaction between treatment groups and *HLA-DQ* genetic backgrounds on VH:CrD and epithelial response to IFNγ GSZ score at PGC, a two-way ANCOVA was conducted using these values at PGC as the dependent variable, *HLA-DQ* genetic background (G1, G2 and G3 genotype groups) and treatment (placebo or drug) as independent variables and baseline VH:CrD ratio and epithelial response to IFNγ GSZ score (from the GFD group), respectively, as a covariate. This analysis included 57 participants, with 1 participant from the placebo group excluded due to an unidentified allele type. The study formulated the following two null hypotheses for the two-way ANCOVA analysis: (1) no VH:CrD (epithelial response to IFNγ GSZ) difference at PCG exists between treatment groups (placebo and drug) while accounting for VH:CrD (epithelial response to IFNγ GSZ) at GFD and (2) no VH:CrD (epithelial response to IFNγ GSZ) differences at PCG exist across *HLA-DQ* genetic backgrounds (G1, G2 and G3 genotype groups) controlling for VH:CrD (epithelial response to IFNγ GSZ) at GFD. For the one-way ANCOVA, only participants in the drug group ($n = 34$) were selected. The null hypothesis for this analysis was that there is no significant effect of *HLA-DQ* genetic background (represented by *HLA-DQ* genotype groups) on VH:CrD within the PGCd group, while adjusting for VH:CrD at GFDd. The one-way ANCOVA regression model included VH:CrD at PGCd as the dependent variable, VH:CrD at GFDd as a covariate and *HLA-DQ* genotype group (G1, G2 and G3) as independent variables. The same type of approach was used for VH and CrD values. Post hoc pairwise multiple comparisons using estimated marginal means calculation (also known as least-squares means) were conducted between the drug and placebo groups for the two-way ANCOVA as well as between *HLA-DQ* genotype groups for the one-way ANCOVA. To address multiple testing, the Bonferroni correction was applied to *P* values (total tests performed = 3). Statistical significance was defined as an adjusted *P* value of <0.05.

## Quantitative real-time PCR

Human duodenal organoids ($n = 3$) were treated with 50, 100 or 200 U ml$^{-1}$ IFNγ (Peprotech) and/or 2, 25 and 50 μM ZED1227 (Zedira) for 24 h. Total RNA was isolated using TRIzol Reagent (15596018), following the manufacturer's instructions, and 500 ng was subjected to cDNA synthesis using an iScript cDNA Synthesis kit (Bio-Rad). Real-time PCR reactions were performed with SsoFast EvaGreen Supermix (1708890, Bio-Rad) and oligonucleotides for human *TGM2* (forward: 5′-TGTGGCACCAAGTACCTGCTCA-3′; reverse; 5′-GCACCTTGATGA GGTTGGACTC-3′) and *GAPDH* (forward: 5′-GTCTCCTCTGACTTC AACAGCG-3′; reverse: 5′-ACCACCCTGTTGCTGTAGCCAA-3′) in triplicate. The results presented were calculated as fold change to the reference sample (nontreated sample), normalized by housekeeping gene expression (*GAPDH*) as described in Schmittgen and Livak[73]. Plot whiskers represent the standard error for mean difference between three independent means.

## Transglutaminase activity assays in Caco-2 cells

Transglutaminase activity was measured using a hydroxamate-based colorimetric method modified from Folk and Cole[74]. In short, each reaction contained 75 mM hydroxylammonium chloride, 30 mM Z-Gln-Gly, 10 mM CaCl$_2$ and 10 mM DTT in 200 mM Tris-HCl buffer (pH 8.0) mixed with cell lysate in a final volume of 100 μl. After a 2-h incubation at 37 °C, the reaction was stopped by the addition of 50 μl of stop buffer (1.67% (wt/vol) FeCl$_3$, 4% (wt/vol) trichloroacetic acid and 4% (vol/vol) HCl). The reaction output was measured at 530 nm, and the activity was expressed as nanomoles of hydroxamate produced in 120 min per milligram of total protein, using L-glutamic acid γ-monohydroxamate for the standard curve.

## HLA genotyping

Five participants had too low coverage either at the *HLA-DQB1* or *HLA-DQA1* locus according to RNA sequencing; thus, their allele typing was not performed. For four of those individuals, blood pellet samples stored at −80 °C were available. DNA was extracted from 100 μl of sample using a QIAamp DNA Blood Mini kit (51104, Qiagen) following the manufacturer's protocol. *HLA-DQB1* and *HLA-DQA1* typing was performed at the Immunogenetics Laboratory at the University of Turku, and the method was based on an asymmetrical PCR and a subsequent hybridization of allele-specific probes, as previously described[75,76].

## Molecular histomorphometry regression model

A regression model predicting VH:CrD ratios, developed in our previous study[13], was used on the current dataset. Models were evaluated by observed versus predicted regression.

## Reporting summary

Further information on research design is available in the Nature Portfolio Reporting Summary linked to this article.

## Data availability

Bulk RNA-sequencing data from participant biopsies and patient-derived intestinal organoids described in this study are available in the European Genome–Phenome Archive under accession numbers EGAS50000000337 and EGAS50000000338. Additional data used in this paper include a full single-cell RNA-sequencing dataset of intestinal regions of adult donors (https://www.gutcellatlas.org/), lists of human duodenal cell types and transporter genes expressed along the upper gastrointestinal tract downloaded from supplementary files included within Busslinger et al.[22], lists of immune cell marker genes downloaded from supplementary files included within Atlasy et al.[58] and pathway gene sets (Reactome, KEGG and BIOCARTA) downloaded from the Human MSigDB Collections at https://www.gsea-msigdb.org/gsea/msigdb/collections.jsp. Source data are provided with this paper. All other data are present in the article and Supplementary Information or are available from the corresponding author upon reasonable request.

## Code availability

Code used in this study is freely available on GitHub at https://github.com/IntestinalSignallingAndEpigeneticsLab/Dotsenko-et-al.-2024.

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

## Acknowledgements

We thank the individuals who participated for making this study possible. We also thank the expert staff for their participation in sample collection. We thank K.-L. Kolho for providing intestinal biopsies to initiate organoid cultures. This work was Dr. Falk Pharma-sponsored clinical trial supported by the Academy of Finland (310011), the Finnish Cultural Foundation, Mary och Georg C. Ehrnrooths Stiftelse, Päivikki and Sakari Sohlberg Foundation, Laboratoriolääketieteen Edistämissäätiö sr and the Competitive State Research Financing of the Expert Responsibility Area of Tampere University Hospital grant. V.D. was supported by the Finnish Cultural Foundation. D.S. received project related support from the German Research Foundation (DFG) Collaborative Research Center SFB TR355/1 (490846870) project B08 (Treg in celiac disease). The funding sources played no role in the design or execution of this study or in the analysis and interpretation of the data. We acknowledge the Adult Stem Cell Organoid Facility from Tampere University for their service. Ethics approvals TUKIJA dnro 223/06.00.01/2017 and EudraCT 2017-002241-30 were obtained for the Dr. Falk Pharma-funded clinical trial. The study was conducted with deidentified data of the participants who had consented to the use of their anonymized data in research. The protocol to initiate human intestinal organoid cultures from biopsies was approved by the Ethics Committee of Tampere University Hospital (ETL code R18082).

## Author contributions

V.D., K.V. and M.M. conceptualized the study. K.V. and V.D. drafted the manuscript. V.D. and K.V. performed data analysis and figure generation. H.H. assisted in statistical analyses. P.H. performed gastroscopies to obtain duodenal biopsies and organoids. B.T., M.H., R.P., J.I., J.T., A.P., J.S., T.Z., R.M., R.G., K.E.A.L. and D.S. assisted in the logistics of data collection and results interpretation. All authors read and approved the final paper.

## Competing interests

V.D. and K.V. received funding from Dr. Falk Pharma to Tampere University to conduct the study. B.T., T.Z., R.M. and R.G. are employees of Dr. Falk Pharma. The data presented here are the subject of patent applications EP24173619.8 and EP24173615.6 filed by Dr. Falk Pharma, and B.T., T.Z., R.M., R.G., V.D. and K.V. are inventors on these applications. M.H. and R.P. are employees of Zedira. A.P. is a consultant for JiLab Oy. J.T. is a consultant for Jilab Oy and Dr. Falk Pharma. K.E.A.L. is a consultant for Amyra, Bioniz Pharmaceuticals, Chugai Pharmaceutical, Dr. Falk Pharma, Itrexon Actobios, TOPAS Therapeutics and Takeda California. D.S. is the data and safety monitor for Boehringer Ingelheim (Phil.) and is a consultant for the Dr. Falk Pharma, Takeda, Immunic, Sanofi and TOPAS Therapeutics. J.I. is the owner of Jilab Oy. M.M. is the founder, owner and Chair of the Board of Maki HealthTech (MHT). MHT receives Management/Advisory Affiliation fees from Dr. Falk Pharma and other funding not related to the research from Topas Therapeutics, Calypso Biotech, Vaccitech, ImmunogenX, Equillium and Immunic. MHT holds patents (patent number 7361480 (United States) and European Patent Office Number 1390753) licensed to Labsystems Diagnostics from where MHT receives royalties via Tampere University Hospital. All other authors declare no competing interests.

## Additional information

**Extended data** is available for this paper at https://doi.org/10.1038/s41590-024-01867-0.

**Correspondence and requests for materials** should be addressed to Keijo Viiri.

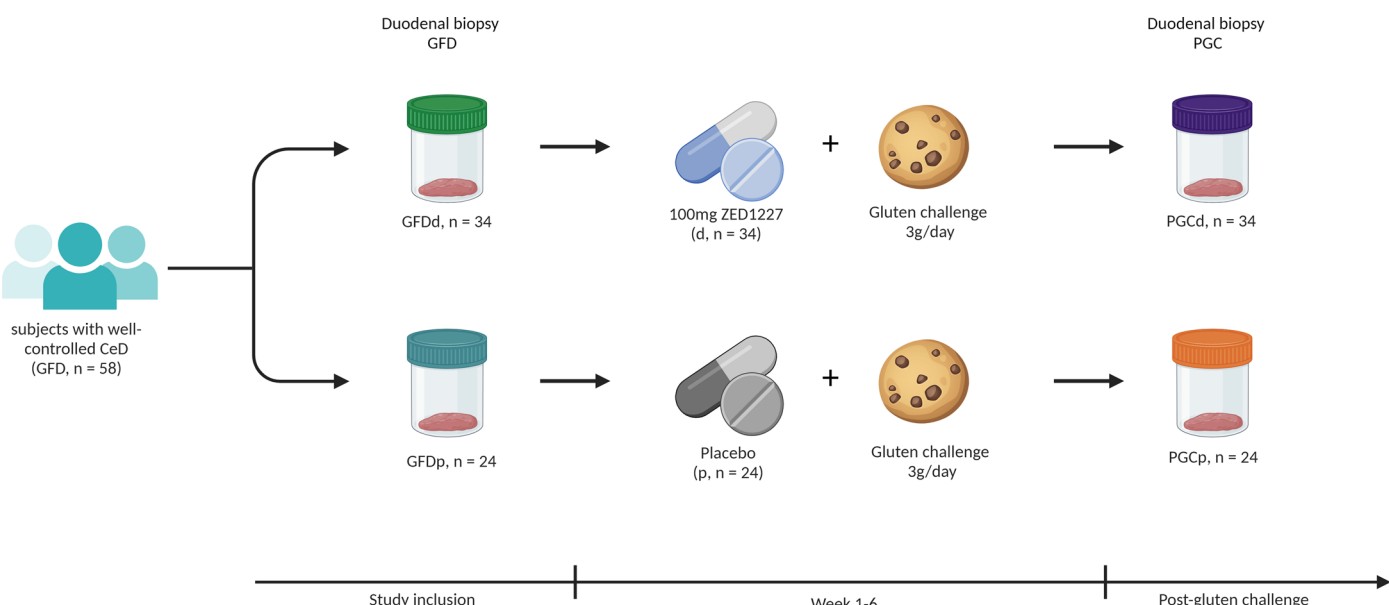

**Extended Data Fig. 1 | Schematic presentation of the study.** Samples ($n = 116$; $n$ of patients = 58), in a form of PAXgene fixed and paraffin-embedded biopsies, were collected from the trial, aimed at dose-finding, and assessing the efficacy and tolerability of a 6-week treatment with ZED1227 capsules vs. placebo in subjects with well-controlled celiac disease undergoing gluten challenge. Biopsy sampling was performed twice: on study inclusion (GFDd, $n = 34$; GFDp, $n = 24$) and at the final visit (PGCd, $n = 34$; PGCp, $n = 24$). Duodenal forceps biopsies were immersed in PAXgene fixative and processed for paraffin block embedding using a standard formalin-free paraffin-infiltration protocol. Created with BioRender.com.

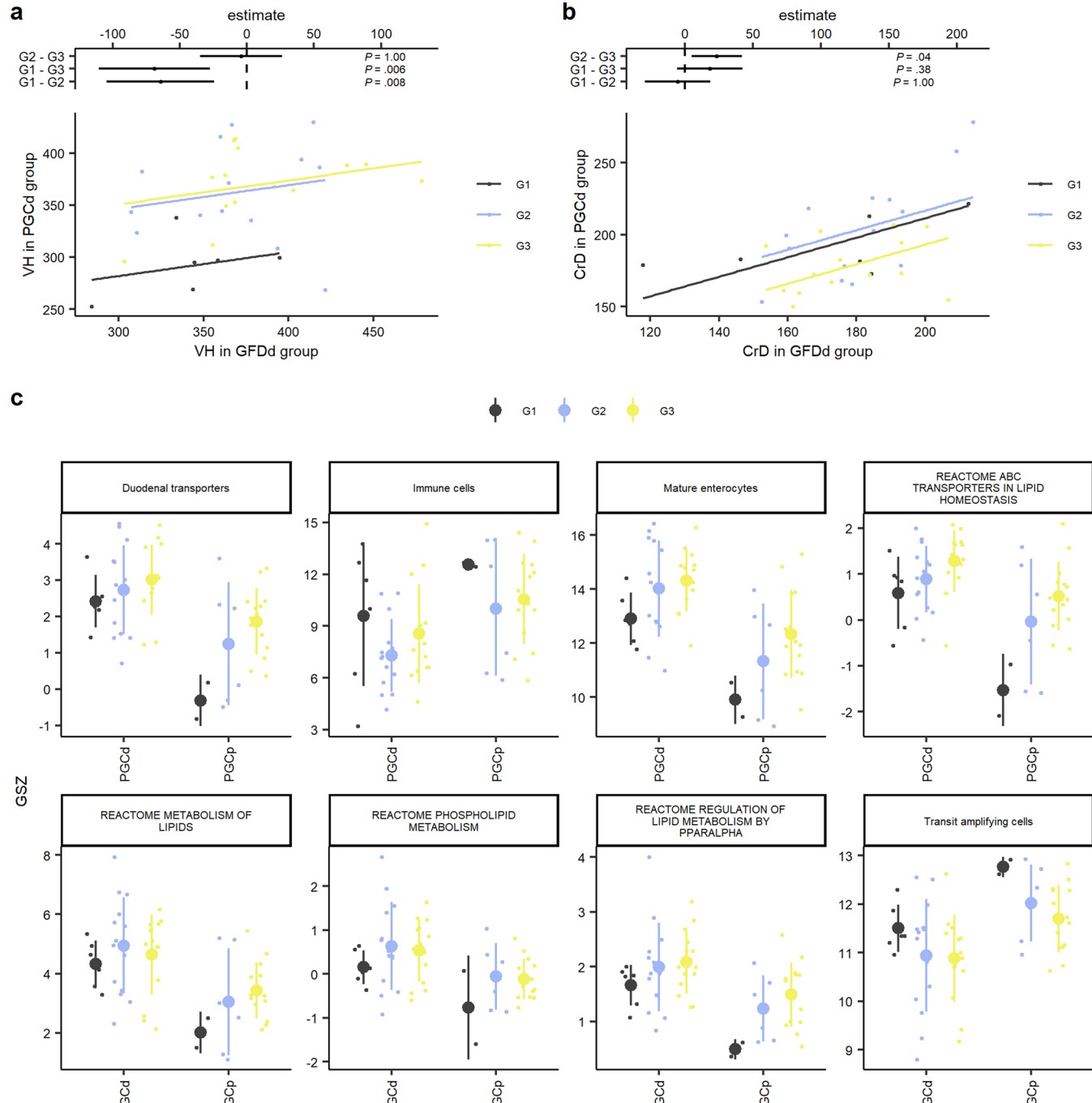

**Extended Data Fig. 2 | Histomorphometric features and molecular pathways displaying reduced control in G1 genotype. a**, Subjects (n = 34), belonging to drug group were selected for one-way ANCOVA. VH at PGCd used as a dependent variable and VH at GFD as covariate and HLA-DQ genotype group (G1, G2, G3) as independent variables. ANCOVA, F (2, 30) = 6.56, P = .004. Post-hoc pairwise multiple comparisons were performed between HLA-DQ genotype groups, with p values Bonferroni adjusted. Results demonstrated as estimate ± 95% CI. **b**, Subjects (n = 34), belonging to drug group were selected for one-way ANCOVA. Cr at PGCd used as a dependent variable and CrD at GFD as covariate and HLA-DQ

genotype group (G1, G2, G3) as independent variables. ANCOVA, F (2, 30) = 3.6, P = .04. Post-hoc pairwise multiple comparisons were performed between HLA-DQ genotype groups, with p values Bonferroni adjusted. Results demonstrated as estimate ± 95% CI. **c**, Gene set Z-score was calculated for gene sets enriched in the categories of transit amplifying cells, mature enterocytes, immune cells, duodenal transporters and Reactome database pathways for patients in drug and placebo groups at PCG (PGCd (n = 34), PGCp (n = 23)). GSZ grouped by HLA-DQ genotype group (G1, G2, G3) and presented as mean (spheres) and sd (vertical lines).

## Extended Data Table 1 | Patient characteristics

| Characteristic | Drug (n=34) | Placebo (n=24) |
|---|---|---|
| Age — yr, mean±sd | 40.7±15.1 | 43.2±14.9 |
| Female sex — n (%) | 22 (64.7) | 17 (70.8) |
| HLA-DQ2 — n (%) | 24 (70.6) | 20 (83.3) |
| HLA-DQ8 — n (%) | 7 (20.6) | 1 (4.2) |
| HLA-DQ2 + HLA-DQ8 — n (%) | 3 (8.8) | 3 (12.5) |
| TG2 IgA —kU/L, median(Q1-Q3) | | |
| GFD | 1.0 (1-2) | 1.0 (1-2) |
| PGC | 1.0 (1-2) | 1.5 (1-7) |
| Ratio of villus height to crypt depth (VH:CrD), mean±sd | | |
| GFD | 2.11±0.34 | 1.95±0.36 |
| PGC | 1.89±0.40 | 1.35±0.65 |

**Extended Data Table 1 | Patient characteristics**

# Reporting Summary

## Statistics

For all statistical analyses, confirm that the following items are present in the figure legend, table legend, main text, or Methods section.

| n/a | Confirmed | |
|---|---|---|
| ☐ | ☒ | The exact sample size (*n*) for each experimental group/condition, given as a discrete number and unit of measurement |
| ☒ | ☐ | A statement on whether measurements were taken from distinct samples or whether the same sample was measured repeatedly |
| ☐ | ☒ | The statistical test(s) used AND whether they are one- or two-sided<br>*Only common tests should be described solely by name; describe more complex techniques in the Methods section.* |
| ☐ | ☒ | A description of all covariates tested |
| ☐ | ☒ | A description of any assumptions or corrections, such as tests of normality and adjustment for multiple comparisons |
| ☐ | ☒ | A full description of the statistical parameters including central tendency (e.g. means) or other basic estimates (e.g. regression coefficient) AND variation (e.g. standard deviation) or associated estimates of uncertainty (e.g. confidence intervals) |
| ☐ | ☒ | For null hypothesis testing, the test statistic (e.g. *F*, *t*, *r*) with confidence intervals, effect sizes, degrees of freedom and *P* value noted<br>*Give P values as exact values whenever suitable.* |
| ☒ | ☐ | For Bayesian analysis, information on the choice of priors and Markov chain Monte Carlo settings |
| ☒ | ☐ | For hierarchical and complex designs, identification of the appropriate level for tests and full reporting of outcomes |
| ☐ | ☒ | Estimates of effect sizes (e.g. Cohen's *d*, Pearson's *r*), indicating how they were calculated |

*Our web collection on statistics for biologists contains articles on many of the points above.*

## Software and code

Policy information about availability of computer code

| Data collection | Duodenal forceps biopsies were collected during clinical drug trial at clinical sites and immersed in PAXgene fixative and processed for paraffin block embedding using a standard formalin-free paraffin-infiltration protocol. Total RNA was extracted from the PaxFPE biopsy specimens using an RNeasy Kit (Qiagen, Hilden, Germany) according to the manufacturer's instructions. Library preparation and transcriptomic data generation was performed by Qiagen NGS Service (Qiagen, Hilden, Germany).  Raw data was de-multiplexed and FASTQ files for each sample were generated using the bcl2fastq2 software version 2.20 (Illumina inc.).<br>RNA from duodenal organoids was isolated using RNeasy Kit (Qiagen, Hilden, Germany) by following manufacturer's instructions.  Preparation of RNA library and transcriptome sequencing was conducted by Novogene Co., LTD (Cambridge, UK).<br>Data quality was checked using the FastQC version v0.11.9. (Cambridge, UK).  3' adapter sequences were trimmed, reads without adapters were kept and reads with <15 bp were removed. Reads were aligned to the human GRCh38 genome using splice aware aligner STAR version 2.7.6 (New York, US) |
|---|---|
| Data analysis | R version 4.3.0 (R Foundation for Statistical Computing, Vienna, Austria). Code is deposited at (https://github.com/ IntestinalSignallingAndEpigeneticsLab/Dotsenko-et-al.-2024). |

For manuscripts utilizing custom algorithms or software that are central to the research but not yet described in published literature, software must be made available to editors and reviewers. We strongly encourage code deposition in a community repository (e.g. GitHub). See the Nature Portfolio guidelines for submitting code & software for further information.

## Data

Policy information about <u>availability of data</u>

All manuscripts must include a <u>data availability statement</u>. This statement should provide the following information, where applicable:
- Accession codes, unique identifiers, or web links for publicly available datasets
- A description of any restrictions on data availability
- For clinical datasets or third party data, please ensure that the statement adheres to our <u>policy</u>

Bulk RNA-sequencing data from patient biopsies and patient-derived intestinal organoids described in this study are available in the European Genome-phenome Bulk RNA-sequencing data from patient biopsies and patient-derived intestinal organoids described in this study are available in the European Genome-phenome Archive (EGA) under accession number EGA50000000324. Additional data used in this paper includes Full single cell RNA-seq dataset intestinal regions of adult donors (https://www.gutcellatlas.org/), lists of human duodenal cell types and transporter genes expressed along the upper gastrointestinal tract downloaded from supplementary files included with Busslinger et al. paper (https://www.sciencedirect.com/science/article/pii/S2211124721001339), lists of immune cells marker genes downloaded from supplementary files included with in Atlasy et al., study (https://www.nature.com/articles/s41467-022-32691-5), pathways gene sets used in study downloaded from Human MSigDB Collections at https://www.gsea-msigdb.org/gsea/msigdb/collections.jsp.

## Research involving human participants, their data, or biological material

Policy information about studies with <u>human participants or human data</u>. See also policy information about <u>sex, gender (identity/presentation), and sexual orientation</u> and <u>race, ethnicity and racism</u>.

| | |
|---|---|
| Reporting on sex and gender | The results of the celiac disease clinical drug trial, using the investigational medical product ZED1227, a TG2 inhibitor, is fully reported with supplemental data in Schuppan and Mäki et al., NEJM 2021;385:35-45. In the clinical study sex and gender are taken care of, we attach the original signed trial protocol (confidential, approved by Dr. Falk Pharma). In the present study, we report the results of our celiac expression profiling samples, the gluten challenge study subject biopsy RNASeq results, Zed1227 100 mg vs. placebo drug. Our RNASeq results are here not given separately for males and females, in drug vs placebo (see Extended Data Table 1, patient characteristics as to female %), as we have no indication in the celiac disease literature that gluten-dependent gene transcript up or downregulations at the duodenal mucosal level could be different in males and females. |
| Reporting on race, ethnicity, or other socially relevant groupings | All Caucasians, no socially relevant groupings were made |
| Population characteristics | Characteristics of the population used in this study is described in the Extended Data Table 1 "Patient characteristics" |
| Recruitment | The published drug trial participant recruitment is not copied to this manuscript, we see it is not relevant to repeat it, we cite the NEJM study. |
| Ethics oversight | We conducted the clinical trial published in NEJM (above) at 20 sites in seven countries (Estonia, Finland, Germany, Ireland, Lithuania, Norway, and Switzerland). The trial was approved by an independent ethics committee at each site. Written informed consent was obtained from each patient before screening. Ethics approvals and informed consents included the pre-specified optional biopsy samples centralized to Tampere for further academic studies (mRNA), the present study. In Finland the protocol was approved by TUKIJA dnro 223/06.00.01/2017, EudraCT 2017-002241-30. For human organoid cultures the protocol was approved by the Ethics Committee of the Tampere University Hospital, Tampere, Finland (ETL-code R18082). |

Note that full information on the approval of the study protocol must also be provided in the manuscript.

# Field-specific reporting

Please select the one below that is the best fit for your research. If you are not sure, read the appropriate sections before making your selection.

☒ Life sciences ☐ Behavioural & social sciences ☐ Ecological, evolutionary & environmental sciences

For a reference copy of the document with all sections, see nature.com/documents/nr-reporting-summary-flat.pdf

# Life sciences study design

All studies must disclose on these points even when the disclosure is negative.

| | |
|---|---|
| Sample size | Sample size was not predetermined statistically. Participant recruitment and sample collection was done in clinical trial reported in Schuppan and Mäki et al., NEJM 2021;385:35-45. For the current study, we used all available samples from ZED1227 100 mg and placebo arms of the trial. |
| Data exclusions | Exclusion criteria were not pre-established. One sample was excluded from secondary differential expression analysis, as it had low total reads. The mean of Total reads for all the samples is 3.51 ± 0.07 million reads. Excluded sample achieved only 1.13 million reads. That sample |

| Replication | is excluded from analyses and all subsequent calculations are performed on 115 left samples. |
| --- | --- |
| Replication | Findings of this study are restricted to the studied cohorts and replication studies were not possible to include. We obtained RNA extracted from PaxFPE biopsy specimens from one single randomized double-blind placebo-controlled clinical drug trial, CEC-3/CEL, conducted from May 16, 2018, to February 27, 2020 and published in NEJM 2021 by Schuppan & Mäki et al. |
| Randomization | Data collection and randomization was performed prior to current study as described in Schuppan and Mäki et al., NEJM 2021;385:35-45. |
| Blinding | Schuppan & Mäki et al. drug trial published in NEJM 2021 was a randomized double-blind gluten challenge trial. For the present study we received the biopsies of the placebo and the ZED1227 100 mg drug arms. All RNA-Seq studies were run in parallel and at the same time. To be able to give results as to baseline and post gluten challenge in placebo vs. drug arm, the trial sponsor provided us with the codes for blinding. |

# Behavioural & social sciences study design

All studies must disclose on these points even when the disclosure is negative.

| Study description | *Briefly describe the study type including whether data are quantitative, qualitative, or mixed-methods (e.g. qualitative cross-sectional, quantitative experimental, mixed-methods case study).* |
| --- | --- |
| Research sample | *State the research sample (e.g. Harvard university undergraduates, villagers in rural India) and provide relevant demographic information (e.g. age, sex) and indicate whether the sample is representative. Provide a rationale for the study sample chosen. For studies involving existing datasets, please describe the dataset and source.* |
| Sampling strategy | *Describe the sampling procedure (e.g. random, snowball, stratified, convenience). Describe the statistical methods that were used to predetermine sample size OR if no sample-size calculation was performed, describe how sample sizes were chosen and provide a rationale for why these sample sizes are sufficient. For qualitative data, please indicate whether data saturation was considered, and what criteria were used to decide that no further sampling was needed.* |
| Data collection | *Provide details about the data collection procedure, including the instruments or devices used to record the data (e.g. pen and paper, computer, eye tracker, video or audio equipment) whether anyone was present besides the participant(s) and the researcher, and whether the researcher was blind to experimental condition and/or the study hypothesis during data collection.* |
| Timing | *Indicate the start and stop dates of data collection. If there is a gap between collection periods, state the dates for each sample cohort.* |
| Data exclusions | *If no data were excluded from the analyses, state so OR if data were excluded, provide the exact number of exclusions and the rationale behind them, indicating whether exclusion criteria were pre-established.* |
| Non-participation | *State how many participants dropped out/declined participation and the reason(s) given OR provide response rate OR state that no participants dropped out/declined participation.* |
| Randomization | *If participants were not allocated into experimental groups, state so OR describe how participants were allocated to groups, and if allocation was not random, describe how covariates were controlled.* |

# Ecological, evolutionary & environmental sciences study design

All studies must disclose on these points even when the disclosure is negative.

| Study description | *Briefly describe the study. For quantitative data include treatment factors and interactions, design structure (e.g. factorial, nested, hierarchical), nature and number of experimental units and replicates.* |
| --- | --- |
| Research sample | *Describe the research sample (e.g. a group of tagged Passer domesticus, all Stenocereus thurberi within Organ Pipe Cactus National Monument), and provide a rationale for the sample choice. When relevant, describe the organism taxa, source, sex, age range and any manipulations. State what population the sample is meant to represent when applicable. For studies involving existing datasets, describe the data and its source.* |
| Sampling strategy | *Note the sampling procedure. Describe the statistical methods that were used to predetermine sample size OR if no sample-size calculation was performed, describe how sample sizes were chosen and provide a rationale for why these sample sizes are sufficient.* |
| Data collection | *Describe the data collection procedure, including who recorded the data and how.* |
| Timing and spatial scale | *Indicate the start and stop dates of data collection, noting the frequency and periodicity of sampling and providing a rationale for these choices. If there is a gap between collection periods, state the dates for each sample cohort. Specify the spatial scale from which the data are taken* |
| Data exclusions | *If no data were excluded from the analyses, state so OR if data were excluded, describe the exclusions and the rationale behind them, indicating whether exclusion criteria were pre-established.* |
| Reproducibility | *Describe the measures taken to verify the reproducibility of experimental findings. For each experiment, note whether any attempts to repeat the experiment failed OR state that all attempts to repeat the experiment were successful.* |

| Randomization | *Describe how samples/organisms/participants were allocated into groups. If allocation was not random, describe how covariates were controlled. If this is not relevant to your study, explain why.* |
|---|---|
| Blinding | *Describe the extent of blinding used during data acquisition and analysis. If blinding was not possible, describe why OR explain why blinding was not relevant to your study.* |

**Did the study involve field work?** ☐ Yes ☐ No

## Field work, collection and transport

| Field conditions | *Describe the study conditions for field work, providing relevant parameters (e.g. temperature, rainfall).* |
|---|---|
| Location | *State the location of the sampling or experiment, providing relevant parameters (e.g. latitude and longitude, elevation, water depth).* |
| Access & import/export | *Describe the efforts you have made to access habitats and to collect and import/export your samples in a responsible manner and in compliance with local, national and international laws, noting any permits that were obtained (give the name of the issuing authority, the date of issue, and any identifying information).* |
| Disturbance | *Describe any disturbance caused by the study and how it was minimized.* |

# Reporting for specific materials, systems and methods

We require information from authors about some types of materials, experimental systems and methods used in many studies. Here, indicate whether each material, system or method listed is relevant to your study. If you are not sure if a list item applies to your research, read the appropriate section before selecting a response.

### Materials & experimental systems

| n/a | Involved in the study |
|---|---|
| ☐ | ☐ Antibodies |
| ☐ | ☒ Eukaryotic cell lines |
| ☐ | ☐ Palaeontology and archaeology |
| ☐ | ☐ Animals and other organisms |
| ☐ | ☒ Clinical data |
| ☐ | ☐ Dual use research of concern |
| ☐ | ☐ Plants |

### Methods

| n/a | Involved in the study |
|---|---|
| ☐ | ☐ ChIP-seq |
| ☐ | ☐ Flow cytometry |
| ☐ | ☐ MRI-based neuroimaging |

## Antibodies

| Antibodies used | *Describe all antibodies used in the study; as applicable, provide supplier name, catalog number, clone name, and lot number.* |
|---|---|
| Validation | *Describe the validation of each primary antibody for the species and application, noting any validation statements on the manufacturer's website, relevant citations, antibody profiles in online databases, or data provided in the manuscript.* |

## Eukaryotic cell lines

Policy information about cell lines and Sex and Gender in Research

| Cell line source(s) | Caco-2 cells (ATCC, Manassas, USA) |
|---|---|
| Authentication | none |
| Mycoplasma contamination | negative |
| Commonly misidentified lines (See ICLAC register) | No commonly misidentified cell lines were used |

## Palaeontology and Archaeology

| Specimen provenance | *Provide provenance information for specimens and describe permits that were obtained for the work (including the name of the issuing authority, the date of issue, and any identifying information). Permits should encompass collection and, where applicable,* |
|---|---|

*export.*

Specimen deposition | *Indicate where the specimens have been deposited to permit free access by other researchers.*

Dating methods | *If new dates are provided, describe how they were obtained (e.g. collection, storage, sample pretreatment and measurement), where they were obtained (i.e. lab name), the calibration program and the protocol for quality assurance OR state that no new dates are provided.*

☐ Tick this box to confirm that the raw and calibrated dates are available in the paper or in Supplementary Information.

Ethics oversight | *Identify the organization(s) that approved or provided guidance on the study protocol, OR state that no ethical approval or guidance was required and explain why not.*

Note that full information on the approval of the study protocol must also be provided in the manuscript.

# Animals and other research organisms

Policy information about studies involving animals; ARRIVE guidelines recommended for reporting animal research, and Sex and Gender in Research

Laboratory animals | *For laboratory animals, report species, strain and age OR state that the study did not involve laboratory animals.*

Wild animals | *Provide details on animals observed in or captured in the field; report species and age where possible. Describe how animals were caught and transported and what happened to captive animals after the study (if killed, explain why and describe method; if released, say where and when) OR state that the study did not involve wild animals.*

Reporting on sex | *Indicate if findings apply to only one sex; describe whether sex was considered in study design, methods used for assigning sex. Provide data disaggregated for sex where this information has been collected in the source data as appropriate; provide overall numbers in this Reporting Summary. Please state if this information has not been collected. Report sex-based analyses where performed, justify reasons for lack of sex-based analysis.*

Field-collected samples | *For laboratory work with field-collected samples, describe all relevant parameters such as housing, maintenance, temperature, photoperiod and end-of-experiment protocol OR state that the study did not involve samples collected from the field.*

Ethics oversight | *Identify the organization(s) that approved or provided guidance on the study protocol, OR state that no ethical approval or guidance was required and explain why not.*

Note that full information on the approval of the study protocol must also be provided in the manuscript.

# Clinical data

Policy information about clinical studies

All manuscripts should comply with the ICMJE guidelines for publication of clinical research and a completed CONSORT checklist must be included with all submissions.

Clinical trial registration | The published clinical drug trial registration is EudraCT 2017-002241-30 (see NEJM 2021;385:35-45).

Study protocol | The signed drug trial final protocol is attached (confidential, approved by Dr. Falk Pharma)

Data collection | As published in NEJM, our European multicenter trial was conducted from May 16, 2018, to February 27, 2020 and data was collected according to the protocol and the biopsy samples for the present study were centralized to Tampere university where we performed the pre-specified mRNA studies. The randomized, double-blind, and placebo-controlled clinical drug trial CEC-3/CEL, EudraCT No.: 2017-002241-30, published as Schuppan and Mäki et al. in New England Journal of Medicine, 2021;385:35-45, was conducted in 20 sites in 7 countries, at university and public/private hospitals and trial center settings: Estonia (Tartu University Hospital, Tartu), Finland (Aava Kamppi Medical Centre, Helsinki; Clinical Research Services Turku – CRST Oy, Turku; FinnMedi Oy, Clinical Trial Center, Tampere), Germany (Institute of Translational Immunology, University Medical Center of the Johannes Gutenberg University, Mainz; Charité University Medicine Berlin, Campus Benjamin Franklin (CBF), Berlin; University Hospital Erlangen, Erlangen; University Hospital of Jena, Jena; University Medical Center Hamburg-Eppendorf, 1st Department of Medicine, Hamburg; Medical University Hospital Tübingen, Internal Medicine 1, Tübingen; Protestant Hospital Essen Steele, Clinic for Naturopathy and Integrative Medicine, Essen; Hospital of the University of Munich-Grosshadern, Medical Clinic and Out-Patient Clinic II, Munich; Clinic for Integrative Medicine and Naturopathy Social Foundation Bamberg "Klinik am Bruderwald", Bamberg), Ireland (University College Hospital Galway, HRB Clinical Research Facility, National University of Ireland, Galway), Lithuania (Hospital of Lithuanian University of Health Sciences, Kauno klinikos, Department of Gastroenterology, Kaunas), Norway (Oslo University Hospital – Rikshospitalet, Oslo; Gjøvik Hospital, Gjøvik; Akershus University Hospital, Lørenskog), and Switzerland (University Hospital Zurich, Department of Gastroenterology and Hepatology, Zürich; Center of Gastroenterology, The Hirslanden Private Clinic Group, Zürich). One site in Austria was initiated but did not enroll any subjects. Chemical laboratory and biopsy samples were centralized for processings and readings and biopsy-extracted RNA was shipped for present studies to us at Tampere University, Faculty of Medicine and Health Technology, Tampere, Finland.

Outcomes | The pre-defined primary and secondary outcome measures of the trial is to be found in the NEJM publication and attached study protocol. We included in the study protocol certain exploratory outcomes to avoid repeating this kind of hugh randomized clinical trial just to get placebo and drug arm biopsy samples during a gluten challenge. It was possible to incorporate this kind of academic study to the industry-sponsored drug trial. Thus, we were able to perform a genome-wide RNASeq on already collected prospective and pre-specified samples. The protocol page 44 says that the biopsies will be used for further immunohistochemistry (IHC) and/or

messenger ribonucleic acid (mRNA) analyses depending on the clinical outcome of the trial (optional investigation). As the clinical outcome published in NEJM was excellent, the trial sponsor, Dr. Falk Pharma, made an agreement with Tampere university and allowed us to proceed with the pre-specified exploratory outcomes/optional samples, and we performed the present study.

# Dual use research of concern

Policy information about dual use research of concern

## Hazards

Could the accidental, deliberate or reckless misuse of agents or technologies generated in the work, or the application of information presented in the manuscript, pose a threat to:

No | Yes

☒ ☐ Public health

☒ ☐ National security

☒ ☐ Crops and/or livestock

☒ ☐ Ecosystems

☒ ☐ Any other significant area

## Experiments of concern

Does the work involve any of these experiments of concern:

No | Yes

☒ ☐ Demonstrate how to render a vaccine ineffective

☒ ☐ Confer resistance to therapeutically useful antibiotics or antiviral agents

☒ ☐ Enhance the virulence of a pathogen or render a nonpathogen virulent

☒ ☐ Increase transmissibility of a pathogen

☒ ☐ Alter the host range of a pathogen

☒ ☐ Enable evasion of diagnostic/detection modalities

☒ ☐ Enable the weaponization of a biological agent or toxin

☒ ☐ Any other potentially harmful combination of experiments and agents

# Plants

| | |
|---|---|
| Seed stocks | *Report on the source of all seed stocks or other plant material used. If applicable, state the seed stock centre and catalogue number. If plant specimens were collected from the field, describe the collection location, date and sampling procedures.* |
| Novel plant genotypes | *Describe the methods by which all novel plant genotypes were produced. This includes those generated by transgenic approaches, gene editing, chemical/radiation-based mutagenesis and hybridization. For transgenic lines, describe the transformation method, the number of independent lines analyzed and the generation upon which experiments were performed. For gene-edited lines, describe the editor used, the endogenous sequence targeted for editing, the targeting guide RNA sequence (if applicable) and how the editor was applied.* |
| Authentication | *Describe any authentication procedures for each seed stock used or novel genotype generated. Describe any experiments used to assess the effect of a mutation and, where applicable, how potential secondary effects (e.g. second site T-DNA insertions, mosiacism, off-target gene editing) were examined.* |

# ChIP-seq

## Data deposition

☐ Confirm that both raw and final processed data have been deposited in a public database such as GEO.

☐ Confirm that you have deposited or provided access to graph files (e.g. BED files) for the called peaks.

| | |
|---|---|
| Data access links<br>*May remain private before publication.* | *For "Initial submission" or "Revised version" documents, provide reviewer access links. For your "Final submission" document, provide a link to the deposited data.* |
| Files in database submission | *Provide a list of all files available in the database submission.* |
| Genome browser session<br>(e.g. UCSC) | *Provide a link to an anonymized genome browser session for "Initial submission" and "Revised version" documents only, to enable peer review. Write "no longer applicable" for "Final submission" documents.* |

## Methodology

| | |
|---|---|
| Replicates | *Describe the experimental replicates, specifying number, type and replicate agreement.* |
| Sequencing depth | *Describe the sequencing depth for each experiment, providing the total number of reads, uniquely mapped reads, length of reads and whether they were paired- or single-end.* |
| Antibodies | *Describe the antibodies used for the ChIP-seq experiments; as applicable, provide supplier name, catalog number, clone name, and lot number.* |
| Peak calling parameters | *Specify the command line program and parameters used for read mapping and peak calling, including the ChIP, control and index files used.* |
| Data quality | *Describe the methods used to ensure data quality in full detail, including how many peaks are at FDR 5% and above 5-fold enrichment.* |
| Software | *Describe the software used to collect and analyze the ChIP-seq data. For custom code that has been deposited into a community repository, provide accession details.* |

# Flow Cytometry

## Plots

Confirm that:

☐ The axis labels state the marker and fluorochrome used (e.g. CD4-FITC).

☐ The axis scales are clearly visible. Include numbers along axes only for bottom left plot of group (a 'group' is an analysis of identical markers).

☐ All plots are contour plots with outliers or pseudocolor plots.

☐ A numerical value for number of cells or percentage (with statistics) is provided.

## Methodology

| | |
|---|---|
| Sample preparation | *Describe the sample preparation, detailing the biological source of the cells and any tissue processing steps used.* |
| Instrument | *Identify the instrument used for data collection, specifying make and model number.* |
| Software | *Describe the software used to collect and analyze the flow cytometry data. For custom code that has been deposited into a community repository, provide accession details.* |
| Cell population abundance | *Describe the abundance of the relevant cell populations within post-sort fractions, providing details on the purity of the samples and how it was determined.* |
| Gating strategy | *Describe the gating strategy used for all relevant experiments, specifying the preliminary FSC/SSC gates of the starting cell population, indicating where boundaries between "positive" and "negative" staining cell populations are defined.* |

☐ Tick this box to confirm that a figure exemplifying the gating strategy is provided in the Supplementary Information.

# Magnetic resonance imaging

## Experimental design

| | |
|---|---|
| Design type | *Indicate task or resting state; event-related or block design.* |
| Design specifications | *Specify the number of blocks, trials or experimental units per session and/or subject, and specify the length of each trial or block (if trials are blocked) and interval between trials.* |
| Behavioral performance measures | *State number and/or type of variables recorded (e.g. correct button press, response time) and what statistics were used to establish that the subjects were performing the task as expected (e.g. mean, range, and/or standard deviation across subjects).* |

## Acquisition

**Imaging type(s)**
Specify: functional, structural, diffusion, perfusion.

**Field strength**
Specify in Tesla

**Sequence & imaging parameters**
Specify the pulse sequence type (gradient echo, spin echo, etc.), imaging type (EPI, spiral, etc.), field of view, matrix size, slice thickness, orientation and TE/TR/flip angle.

**Area of acquisition**
State whether a whole brain scan was used OR define the area of acquisition, describing how the region was determined.

**Diffusion MRI** ☐ Used ☐ Not used

## Preprocessing

**Preprocessing software**
Provide detail on software version and revision number and on specific parameters (model/functions, brain extraction, segmentation, smoothing kernel size, etc.).

**Normalization**
If data were normalized/standardized, describe the approach(es): specify linear or non-linear and define image types used for transformation OR indicate that data were not normalized and explain rationale for lack of normalization.

**Normalization template**
Describe the template used for normalization/transformation, specifying subject space or group standardized space (e.g. original Talairach, MNI305, ICBM152) OR indicate that the data were not normalized.

**Noise and artifact removal**
Describe your procedure(s) for artifact and structured noise removal, specifying motion parameters, tissue signals and physiological signals (heart rate, respiration).

**Volume censoring**
Define your software and/or method and criteria for volume censoring, and state the extent of such censoring.

## Statistical modeling & inference

**Model type and settings**
Specify type (mass univariate, multivariate, RSA, predictive, etc.) and describe essential details of the model at the first and second levels (e.g. fixed, random or mixed effects; drift or auto-correlation).

**Effect(s) tested**
Define precise effect in terms of the task or stimulus conditions instead of psychological concepts and indicate whether ANOVA or factorial designs were used.

**Specify type of analysis:** ☐ Whole brain ☐ ROI-based ☐ Both

**Statistic type for inference**
(See Eklund et al. 2016)
Specify voxel-wise or cluster-wise and report all relevant parameters for cluster-wise methods.

**Correction**
Describe the type of correction and how it is obtained for multiple comparisons (e.g. FWE, FDR, permutation or Monte Carlo).

## Models & analysis

| n/a | Involved in the study |
|-----|----------------------|
| ☐ | ☐ Functional and/or effective connectivity |
| ☐ | ☐ Graph analysis |
| ☐ | ☐ Multivariate modeling or predictive analysis |

**Functional and/or effective connectivity**
Report the measures of dependence used and the model details (e.g. Pearson correlation, partial correlation, mutual information).

**Graph analysis**
Report the dependent variable and connectivity measure, specifying weighted graph or binarized graph, subject- or group-level, and the global and/or node summaries used (e.g. clustering coefficient, efficiency, etc.).

**Multivariate modeling and predictive analysis**
Specify independent variables, features extraction and dimension reduction, model, training and evaluation metrics.

nature portfolio | reporting summary

April 2023

