## [Peer Review File · Nature Immunology]

Peer Review Information

Journal: Nature Immunology

Manuscript Title: Transcriptomic analysis of intestine following administration of a transglutaminase 2 inhibitor to prevent gluten-induced intestinal damage in celiac disease

Corresponding author name(s): Dr Keijo Viiri

Editorial Notes:

Transferred manuscripts This manuscript has been previously reviewed at another journal that is not operating a transparent peer review scheme. This document only contains reviewer comments, rebuttal and decision letters for versions considered at Nature Immunology

Reviewer Comments & Decisions:

Decision Letter, initial version:
--

17th Jul 2023

Dear Dr Viiri,

Thank you for providing a response to reviewers concerns for your article, "Transglutaminase 2 inhibitor protects from gluten-induced intestinal damage in celiac disease – Transcriptomic analysis of a randomized gluten challenge study". While we cannot accept the manuscript in it's current form for publication we would be interested in considering a revised version that addresses the reviewers serious concerns. Please bear in mind that we will be reluctant to approach the referees again the in absence of major revisions. We would also like to request that you pay particular attention to addressing the concerns of Reviewer #4.

If you choose to revise your manuscript taking into account all reviewer and editor comments, please highlight all changes in the manuscript text file [OPTIONAL: in Microsoft Word format].

* If you have not done so already please begin to revise your manuscript so that it conforms to our Article format instructions at <http://www.nature.com/ni/authors/index.html>. Refer also to any guidelines provided in this letter.

The Reporting Summary can be found here:

When submitting the revised version of your manuscript, please pay close attention to our [href="https://www.nature.com/nature-portfolio/editorial-policies/image-integrity">Digital Image Integrity Guidelines](https://www.nature.com/nature-portfolio/editorial-policies/image-integrity). and to the following points below:

[redacted]

If you wish to submit a suitably revised manuscript we would hope to receive it within 6 months. If you cannot send it within this time, please let us know. We will be happy to consider your revision so long as nothing similar has been accepted for publication at Nature Immunology or published elsewhere.

Nature Immunology is committed to improving transparency in authorship. As part of our efforts in this direction, we are now requesting that all authors identified as 'corresponding author' on published papers create and link their Open Researcher and Contributor Identifier (ORCID) with their account on the Manuscript Tracking System (MTS), prior to acceptance. ORCID helps the scientific community achieve unambiguous attribution of all scholarly contributions. You can create and link your ORCID from the home page of the MTS by clicking on 'Modify my Springer Nature account'. For more information please visit please visit www.springernature.com/orcid.

Thank you for the opportunity to review your work.

Sincerely,

Stephanie Houston
Editor
Nature Immunology

Author Rebuttal to Initial comments

See inserted PDF

Please note that all the line numbers refer to non-tracked clean version of the manuscript:

Dotsenko_et_al_manuscript_revised_clean

Reviewer #1:

Remarks to the Author:

The article by Dotsenko and coworkers entitled “Transglutaminase 2 inhibitor protects from gluten-induced intestinal damage in celiac disease-Transcriptomic analysis of a randomized gluten challenge study” evaluates the transcriptional changes taking place in the duodenum of treated celiac disease patients following gluten challenge and administration of the transglutaminase 2 inhibitor, ZED1227, whose has been previously shown to prevents gluten-induced mucosal damage. This manuscript represents a logical extension of their work and confirms the impact of TG2 inhibition on mucosal healing. In addition, capitalizing on the samples collected and stored during the Phase I clinical trial, the authors demonstrate that administration of ZED1227 alters gene expression associated with IFN-g signaling and epithelial IFN-g response and that the HLA genotype explains the variations in the efficacy of ZED1227. Overall, the results support the conclusion that the major impact of TG2 inhibition is on the IFN-g pathway, whose activation is critical for the induction of villous atrophy in celiac disease, and that ZED1227 is less efficient in controlling immune responses driving villous atrophy in patients with the G1 genotype (high gluten-response group). Overall the manuscript merits publication after some questions/issues have been addressed

- **RESPONSE:** We thank reviewer for her/his valuable comments and hope we have addressed them all appropriately

Specific comments:

- The abstract mentions that “ZED1227 treatment preserved transcriptome signatures associated with mucosal morphology, inflammation, cell differentiation, and nutrient absorption to the level of the GFD group” but this not thoroughly shown in the manuscript except in Fig. 2. In particular, could the authors be more specific about which pathways are less controlled in the G1 phenotype (Figure 4).

- **RESPONSE:** We acknowledge that conducting further pathway analysis beyond the IFNg signaling pathway is a logical extension of our efforts to identify patients with a diminished response to ZED1227 treatment. GSZ for 4 cell-differentiation-related pathways (as in Fig. 2) were calculated and presented in supplemental data. Moreover, we discovered that PPAR and associated lipid signalling pathways are less controlled in G1 group. These analyses are now shown in Fig. S6C and discussed in the text lines 563 – 578.

- Figure 1E is not informative. Instead, the authors should show the correlation of effect size between the groups, the expectation being that there will be a correlation between the effect size of the comparison PGcP vs GFD and the effect size of the comparison PGcP and PGcd.

- **RESPONSE:** Correlation of log2FC is now added to Fig. 1E and text in lines 335-337 were added “When all detected genes log2FC from PGCp versus GFDp comparison were compared to PGCp versus there was a positive correlation, suggesting similar pattern of expression changes in both groups (Fig. 1E)”.

- Figure 2: Genes shown in Figure 2A should be categorized by modules based on their predictive role. Figure 2D: Details should be given regarding the cellular composition of the different categories of cells identified as being impacted by the treatment. The authors should also include an IEL and epithelial analysis looking at NKG2D, perforin, granzyme, CD94, HLA-E, MICA to determine whether the effector cytotoxic IEL responses is altered.

- **RESPONSE:** For the genes shown in Figure 2A, we have now categorized them based on their gene ontologies. Additionally, we performed deconvolution analysis on our bulk RNA-seq data using published single-cell transcriptomic data to determine the cellular composition. The results plotted and added to the supplemental figure S4. Text is updated in lines 364-368 “Bulk RNA sequencing deconvolution, that used duodenal single cell RNA-sequencing as reference, revealed similar patterns in cell proportion distributions, like decrease in enterocytes numbers accompanied with small increase in stem cells numbers in PGCp group (Figure S4A-B).” We also analyzed the effector cytotoxic IEL responses (judged by marker genes expression) and detected slight increase in HLA-E expression but for the other markers we failed to detect any profound alteration during the gluten challenge (Figure S4C). This probably due to the scarcity of these cell types over massively abundant epithelial cell types.

- For Figure 3: Could the authors also include an analysis for the IL-21 signaling pathway.

- **RESPONSE:** Given that the IL-21 signaling pathway is known to play a role in CD pathogenesis and contribute to the mucosal Th-1 cell response, we agree that including this pathway in the analysis is interesting. We analysed IL-21 signalling pathway and this is now added to Fig. S5B and communicated in lines 397 – 399.

- Figure 3C: Are the 4 patients presenting the highest active epithelial IFN γ response after administration of ZED1227 having the most severe duodenal lesions?

- **RESPONSE:** The four patients in question do indeed exhibit most severe duodenal lesions, as indicated by the measured VH:CrD values. We modified Figure 3C to include these values and it is evident that these four patients fall into quartile of lowest VH:CrD.

- Figure 3F: A dose-response should be shown.

- **RESPONSE:** qPCR results showing TGM2 expression in human duodenal organoids treated with different doses of IFN γ and ZED1227 is now included in Fig. S5C.

- Fig.5 is an extension of the morphometric analysis performed in the first study and is disconnected from the rest of the current manuscript. It should either be presented upfront in Figure 1 or shown in supplementary data.

- **RESPONSE:** Figure 5 may appear disconnected from the rest of the study but on the other hand it is showing the important piece of evidence that ZED1227 improves histomorphometry also at the molecular level. In the first study (Dotsenko et al. 2021) we validated the model with the independent published data set of healthy controls and celiac patients. Admittedly this independent validation was not ideal since our model was built solely on pre- and post-gluten challenged celiac disease patient data. Here we provide independent validation of the model by using identical gluten challenge data. Therefore, we feel that this piece of data deserves to be shown at the end of the manuscript.

- In the discussion, second paragraph, the authors state that ZED1227 results in “protection from villous atrophy and intraepithelial lymphocytosis (Fig. 2D)”. However, the composition and attributes of intraepithelial lymphocytes are not analyzed (see comment for Figure 2). Furthermore, in the discussion the authors should discuss the observation that the treatment is less efficient in controlling villous atrophy in the G1 genotype. Furthermore, their conclusion would gain in impact if they can put forward hypotheses, based on which pathways failed to be controlled (integrate analysis of Figure 2 into figure1), that would explain the lower efficiency of ZED1227 in the G1 genotype (is there for example a link with a lower control of the IFN γ pathway or the IEL cytotoxic phenotype).

- **RESPONSE:** We have discussed the observation that the treatment is less efficient in controlling villous atrophy in the G1 genotype. We also performed molecular analyses and show that G1 group is more pathognomonic when gene set Z-scores for ‘transit amplifying cells’, ‘mature enterocytes’, ‘immune cells’ and ‘duodenal transporters’ are assessed. As discussed above we didn’t really see any profound induction of IEL cytotoxic pathways, except maybe for the HLA-E and GZMA. Moreover, these were not more severely affected in G1 group either (data not shown). Instead, we discovered that PPAR and associated lipid signalling pathways are less controlled in G1 group (Fig. S6C). As IFN γ is known to reduce PPAR signalling it is likely that these are just downstream events of increased IFN γ response in G1 group.

- Table 3: PGCP is spelled PGCb.

- **RESPONSE:** This has been now corrected. We have discussed the observation that the treatment is less efficient in controlling villous atrophy in the G1 genotype.

Reviewer #2:

Remarks to the Author:

This study performs a companion transcriptomic, enteroid and genotype analysis to accompany their New England Journal of Medicine article demonstrating histomorphometric benefit (villus-crypt height, CD3+ infiltration) of a randomized controlled trial of 100 mg daily of ZED1227, and tissue transglutaminase 2 inhibitor. Inclusion for this study included being either DQ2 or DQ8 positive, and being on a gluten free diet (GFD) for one year. Patients were then randomized to either ZED1227 (here, highest dose, n=34), placebo (n=24) with all receiving 3 grams of gluten challenge (PGCd, PGCP)

respectively) for 6 weeks. Bulk RNAseq was performed at baseline (after one year on confirmed GFD) and following randomization.

Major claims of this manuscript include, a) absence of transcriptomic changes in the PGCd vs. GFD (gold standard) compared to PGCp (placebo) vs. GFD groups, b) replication of many of the in vivo findings with IFNg treatment in enteroids, and c) that genotype classes (G1, G2, G3) can be used a priori to predict serial histomorphometric responses to ZED1227 response with gluten challenge.

Strengths of this manuscript include its companion biospecimen sampling with a clinical trial and serial duodenal biopsy analyses. An additional strength of this manuscript is the prior literature and clean mechanism of TG2 activation in celiac disease pathogenesis. This results in very strong biomarkers for testing treatment response (IgA TG2, IFNg response phenocopying epithelial transcriptional responses, villus-crypt ratios)

These strengths are outweighed by substantive weaknesses, most centrally, the presence of only moderate advances past their 2021 clinical trial.

Major limitations

- Inconsistent and confusing accounting of patients between their trial and the present efforts. This may be attributed to their inclusion criteria here, namely DQ2 and DQ8 genotypes. There appears to be non-random imbalance between their PGCd and PGCp groups with respect to the genotypes (Table 1). The trial lists what appears to be 30 PGCp patients, with only 24 listed here. With these relatively modest sample sizes, these differences may have outsize effects on any genotype inferences.

- **RESPONSE:** We thank reviewer for her/his valuable comments, and we try to address them all accordingly. One of the inclusion criteria for our study was Human leukocyte antigen DQ (HLA-DQ) typing compatible with celiac disease, which required patients to be positive for either HLA-DQ2 or HLA-DQ8. These HLA types are present in the vast majority of individuals with celiac disease (>90% according to [DOI: 10.1016/S0198-8859\(03\)00027-2](https://doi.org/10.1016/S0198-8859(03)00027-2)). However, it is important to note that during the randomization process, patient genotypes were not controlled for, resulting in a random distribution of genotypes among the groups. Table 1 in our study represents only patient characteristics. We fully acknowledge that this has led to small sample sizes, particularly in the G1 group. We recognize that this limitation poses a challenge for statistical analysis, and we have explicitly mentioned it in the discussion section of our study lines 579-587. Additionally, it is worth noting that the RNASeq analyses for smaller patient groups were designated as optional and exploratory in our study protocol, contingent upon the results of the CEC-3 clinical trial. Participants in the trial were required to provide separate written informed consent specifically for these exploratory (optional) studies, and not all participants agreed to participate. Consequently, the numbers for these analyses were somewhat smaller due to the varying levels of participant consent.

-- In general, a more complete accounting of all doses, instead of merely the highest dose (100 mg) should have been provided. The presence of dose-response effects would be extremely illuminating and important to show.

- **RESPONSE:** We agree, that including all tested doses could be beneficial for study. But, since the transcriptomic study was optional, RNA isolation was not performed for all drug groups. The rationale behind this decision was to focus on the drug group that exhibited the most significant improvement compared to the placebo group in order to investigate potential transcriptomic changes in that particular group. This approach allowed us to prioritize the analysis and utilize our resources effectively. That is mentioned in line 579, among study limitations.

- Efforts here to show genotype-dependent differences (G1, G2, G3) are hampered by the small sample sizes, with somewhat arbitrary lumping of genotypes. Given the highest pathogenic effects inferred from prior literature with the DQ2.5 allele, it is not surprising that the G1 grouping (includes the DQ2.5 homozygotes) shows the least histomorphometric benefit (Table 3) compared to G2 and G3. Given the small sample sizes, combined with the asymmetric distribution of genotypes (Table 1) between PGCD vs. PGCp, these findings are of only marginal impact and significance. The absence of a replication cohort limits the rigor of this finding.

- **RESPONSE:** We fully acknowledge that the limited sample size poses a limitation for statistical analysis. To address this issue, we performed additional genotyping for the 5 patients listed in Table 2, for whom we were unable to determine genotypes from the initial RNA sequencing. However, this does not change the sizes of G1 group both for drug and placebo group. We mentioned in manuscript that small group sizes may have implications for statistical power and the generalizability of our results (lines 580-582).

- The absence of population diversity, given the presence of celiac disease, limits generalizability of their findings.

- **RESPONSE:** CEC3-3/CEL Clinical trial, from which samples used in this study are originated, included sites in Austria, Estonia, Finland, Germany, Ireland, Lithuania, Norway, and Switzerland. Centers were based in European countries, and the patients included in our study were of Caucasian ethnicity, as indicated in supplemental table 1. According to available research, CeD prevalence is higher among non-Hispanic white populations (DOI: 10.1007/s10620-014-3514-7 and DOI: 10.1038/ajg.2015.8). Additionally, we are not aware of research linking race and ethnicity to CeD pathogenesis. In our view, the disease shares similarities in terms of triggers, clinical presentation, diagnostic criteria, pathogenetic mechanisms, and treatment across different ethnic groups. Therefore, we believe that our findings can be generalized beyond this our study population. Nevertheless, we acknowledge the importance of considering ethnicity as inclusion criteria in future studies.

- The bulk RNASeq findings are somewhat arbitrary in their findings (log2FC) and primarily designed to demonstrate no differences between GFD and PGCD. The authors miss a major opportunity to pair the transcriptomes pre- and post- 6 week randomization, as inter-individual differences undoubtedly contribute a substantial fraction of transcriptional variance. Furthermore, much of the analyses in Fig 1 may obscure findings by including pre- as well as post-randomization transcriptomes altogether (Fig 1f)

- RESPONSE:** During the trial, patients were randomly assigned to either the drug or placebo groups. Biopsies were collected at two time points: baseline (before gluten challenge, with or without drug, referred to as GFD in the manuscript) and at the end of the study (after a 6-week gluten challenge with drug (PGCd) or with placebo (PGCp)). We recognize the paired nature of the samples and have utilized this information to assess transcriptomic changes in the PGCp vs GFDp and PGCd vs GFDd comparisons. We acknowledge that the abbreviation of the baseline group as GFD may have caused confusion, as we did not explicitly state that the PGC samples were compared to themselves at the baseline (so PGCp to GFDp and PGCd to GFDd). We have now tried our best to provide a clearer description in the revised version of the manuscript and changed abbreviations to proper forms where applicable. In addition, schematic presentation of the study is now shown in Fig. S1.

However, as reviewer pointed out, there may be a bias of inter-individual differences present in the PGCp vs PGCd comparison. This is evidenced by the higher number of differentially expressed genes detected in the PGCp vs PGCd comparison (180 genes) compared to the PGCp vs GFDp comparison (95 genes). This disparity can be attributed to the lack of paired nature in the PGCp vs PGCd samples and this has been discussed in the manuscript.

- The IFNG studies with enteroids are not particularly impactful, recreating previously reported pathways. These model systems would not necessarily capture other key cells (e.g. intraepithelial lymphocytes) for which it would be important to understand what effects that transglutaminase might have

- RESPONSE:** Given that the majority of the transcriptomic signal in our biopsies originates from epithelial cells, more than 85%, as indicated by the deconvolution results (Fig. S4A), our primary focus was to investigate the epithelial response to IFNG. We believe that human duodenal enteroids serve as an optimal model system for studying this aspect. However, we acknowledge the concern that our organoid model does not encompass the lymphocytes that are important for celiac disease pathogenesis, but our purpose was namely to study epithelial IFN-gamma response.

- More generally, longer-term effects beyond 6 weeks, at various doses, modulated by varying levels of gluten intake (3 grams tested here) would be more impactful to understand

- RESPONSE:** The CEC3-3/CEL Clinical trial was designed with 6 week 3 grams daily gluten challenge. This was sufficient to cause mucosal damage in placebo patients and provided time window to study the efficacy of ZED1227. Nevertheless, we acknowledge that longer-term studies with varying doses of both gluten intake and ZED1227 treatment, accompanied by larger cohort sizes, are essential for a comprehensive understanding of the treatment's long-term and dose-dependent effects and could be considered as future research.

Reviewer #3:

Remarks to the Author:

A. The authors present results of RNAseq analysis of duodenal biopsies from celiac disease patients before and after a gluten challenge during which they were randomized to the TTG inhibitor ZED1227 or placebo.

B. The results are of high importance as they are providing molecular evidence (as opposed to mere morphologic measurements) to demonstrate the efficacy of the drug compared to GFD alone (baseline samples). Additionally, they validate a previous multiplex gene expression panel to determine Vh: Cd and examined an interferon gamma signature by comparing the gene expression profiles from the patients to profiles from enteroids cultured from patients unaffected by celiac disease.

C. The manuscript would benefit from some polishing of the writing in places, particularly in the third paragraph of results. Overall, there are a lot of non-standard abbreviations which make the manuscript extremely difficult to follow.

- **RESPONSE:** We thank reviewer for her/his valuable comments, and we have now polished the writing and improved the readability of the manuscript. Regarding the abbreviations, we understand that they may appear non-standard. However, we have followed the same abbreviations used in our previous paper published in CMGH Dotsenko et al. 2021. To ensure clarity for readers, we have provided a list of abbreviations in the manuscript.

F. When were biopsies collected? It appears that there were three biopsies, but unclear how baseline and screening relate to “pre-gluten challenge” vs “post-gluten challenge”. What was the comparison between baseline and run-in biopsies?

- **RESPONSE:** During the trial, patients were randomly assigned to either the drug or placebo groups. Biopsies were collected at two time points: baseline (before gluten challenge, with or without drug, referred to as GFD in the manuscript) and at the end of the study (after a 6-week gluten challenge with drug (PGCd) or with placebo (PGCp)). “Baseline” in our context is synonymous with “run-in”, “pre-gluten challenge” and “GFD”. We apologize for any confusion that may have arisen from the sentence in Lines 138-139, which states, “Biopsy sampling was performed at the baseline, the screening period and at post gluten challenge.” This sentence is indeed confusing, and it has been rephrased now at line # as “Biopsy sampling was performed twice: on study inclusion (denoted here as GFD) and at the final visit (denoted here as PGC) (fig. S1).”. Thank you for bringing this to our attention. We have also made a schematic presentation of the study in Fig. S1.

H. When only one gene is differentially expressed, please name it (abstract, results, etc)

- **RESPONSE:** This gene, RABGGTA, is mentioned now in fig.1C. Additionally, the lists of differentially expressed genes in all comparisons is included as supplemental data.

Line 212 – is there an error?

- **RESPONSE:** Full sentence is “Due to low expression, the low resolution (Field1, allele group) was taken into consideration in the subsequent statistical analysis.” The term "field" in this context refers to the hierarchical classification of HLA alleles based on sequence resolution and used in publications, e.g. DOI:10.1007/s00281-021-00901-9.

Reviewer #4:

Remarks to the Author:

I found this paper reasonably clear in terms of the biology and key conclusions, and these are interesting and plausible. However, I struggled to work out exactly what statistical approaches had been used and whether these were appropriate and supported the conclusions of the paper. In particular, the paper needs to more clearly acknowledge that the 3 sample groups are pre and post treatment samples from 2 groups of individuals, so that (eg) the comparison of expression levels in PGCp with GFD needs to allow for the fact that the GFD group contains the baseline measurements of the placebo individuals. I found one line (L 197) saying “paired nature of samples was considered for detection of differentially expressed genes “ but no description of how this was done, or whether the various other tabulated/plotted quantities allow for this. My own preference would be to describe the results entirely in terms of the change from baseline in the two treatment groups; I think this would be much clearer than the current analysis. I would also focus more on the magnitude of change in expression rather than whether it was “significant” or not. There are a number of other places where the statistical methods weren’t completely clear to me (eg, treatment of multiplicity, exactly what is tabulated/plotted). It would be helpful to include a statistical methods section, perhaps as supplementary material.

- **RESPONSE:** We thank reviewer for her/his constructive comments and we will describe below in greater detail how we have addressed these concerns in the revised manuscript.

Specific comments:

L56: Somewhere say a bit more about the data and the trial from which the samples come. Eg, why more subjects in active than control group (trial protocol suggests 1:1:1:1 randomisation between placebo and 3 tmt arms) ? Any dropout/missing data, etc? Where samples not collected for the other tmt arms or just not included in this analysis? In either case, does this missingness potentially lead to bias? Note the protocol suggests an interim analysis was planned, this needs to be allowed for in any analysis of the primary efficacy variable (change in VH:CrD) and possibly gene expression.

- **RESPONSE:** We have now provided more detailed description of the sample collection process in MATERIAL AND METHODS section, Patients and Biopsies subsection, lines 126-151.

This study utilized samples from two groups: placebo and the 100-mg ZED1227 group, which represented the highest dose drug group showing the most significant improvement compared to the placebo group. In total, 58 patients (drug group, n = 34; placebo group, n = 24, total number of biopsies = 116) out of the 68 patients who had sufficient biopsy samples at both timepoints in the trial original were included, as these exploratory (optional) studies required separate written informed consent from the patients. Participation rate was around 85%. Demographic characteristics and duodenal histomorphometry changes in form of VH:CrD of the patients in original cohort and in present study are presented in tables S1 and S2. Though it is hard to assess, to which degree missing 10 pairs of biopsy samples lead to bias, demographic and histomorphometric characteristics were very similar with the original cohort.

The primary endpoint of the CEC-3/CEL trial was the "Attenuation of gluten-induced change in intestinal mucosal morphology," measured morphometrically through biopsy analysis. Therefore, biopsies were collected only at two time points: baseline (referred to as GFD in the manuscript) and week 6 following the continuation of gluten challenge (PGC). Since RNA was isolated from paraffin-embedded biopsies, we were only able to obtain transcriptomic data for these two time points. The interim analysis in the trial focused on patient-reported outcomes and serological markers, which were beyond the scope of this particular study and thus not included in the manuscript.

L63 I assume GFD is everyone at baseline, but should formally define here. Comparing numbers of differentially expressed genes can be fraught because it is so dependent on power, significance thresholds etc. Taking these numbers at face value, we would conclude PGCD has a similar transcriptomic profile to GFD, so why do we see twice as many gene differences comparing PGCP with PGCD than with GFD? This is either interesting or an artifact and I'm not sure which. It would be interesting to consider the magnitude of changes at the DEG

- **RESPONSE:** GFD represents the baseline for all participants. We made changes in the manuscript specifying this, for example line 138 says: "Biopsy sampling was performed twice: on study inclusion (denoted here as GFD) and at the final visit (denoted here as PGC) (fig. S1)."

Line 63 (now lines 67-69) also changed to "Transcriptomic changes were identified in the comparisons between PGCP and GFDp, as well as PGCP and PGCD groups.

However, only one differentially expressed gene was detected in the comparison between PGCd and GFD.”

In the design model, when assessing transcriptomic changes in the PGCp vs GFDp and PGCd vs GFDd comparisons, we took into account the paired nature of the before and after treatment samples. Line 209 now says: “Pre-and post-treatment samples were compared, and the paired nature of samples was included as a term into multi-factor design formula.” However, there are no before and after pairs in the PGCp vs PGCd comparison. Therefore, we suggest that the presence of inter-individual differences leads to a higher number of differentially expressed genes detected in the PGCp vs PGCd comparison (180 genes) compared to the PGCp vs GFD comparison (95 genes) and this has been mentioned in the manuscript.

We acknowledge that the use of the abbreviation "GFD" for the baseline group may have caused confusion, as we did not explicitly state that the PGC samples were compared to themselves at the baseline (so PGCp to GFDp and PGCd to GFDd). We have now tried our best to provide a clearer description in the revised version of the manuscript and changed abbreviations to proper forms where applicable.

- L 140 How many organoids, from how many samples/people?

- **RESPONSE:** Human duodenal organoids were generated from 3 non-celiac disease donors. Changes made in lines 155.

- L 197 “paired nature of samples was considered for detection of differentially expressed genes “---how? Was this a difference in expression between time points?

- **RESPONSE:** Line 197 (now Line 209) now says: “Pre-and post-treatment samples were compared, and the paired nature of samples was included as a term into multi-factor design formula.” We acknowledge that the use of the abbreviation "GFD" for the baseline group may have caused confusion, as we did not explicitly state that the PGC samples were compared to themselves at the GFD (so PGCp to GFDp and PGCd to GFDd). We tried our best to provide a clearer description in the revised version of the manuscript and changed abbreviations to proper forms where applicable.

- L198 “The obtained P values were adjusted for multiple testing using the Benjamini-Hochberg method.” B-H is a method for calculating false discovery rates (FDR)--- it doesn’t in normal application “correct” p-values but produces a different measure of significance which must be interpreted differently. However the rest of the paper quotes p-values—can you clarify?

- **RESPONSE:** We agree that it is necessary to change the term "adjusted p-value" to "FDR" in cases where the Benjamini-Hochberg method was used. We have now modified the manuscript text and captions to explicitly state which method was used in each particular case regarding p-value adjustment. Changes made in line 198 (now line 209): “The obtained P values were adjusted for multiple testing using the Benjamini-Hochberg method. Genes with an FDR < .05 and

absolute log₂-fold change $|\log_2FC| \geq 0.5$ found by DESeq2 were assigned as differentially expressed.” and pictures captions where applicable.

- Table 1 Add a measure of variability to these quantities. What is DQ2 + Dq8? How can this be 12.5% if DQ8 is 4.2%? I think I would prefer counts to % here.

- **RESPONSE:** DQ2+DQ8 indicates that the patient carries both types of high-risk haplotypes for celiac disease. This is distinct from being HLA-DQ8 positive, which means only the DQ8 haplotype is present. We have modified the table 1 and specified the patient counts. Changes made in Table 1, variances added, and counts accompanied with %.

- L267 “virtually” -> “clearly”? How do these plots account for the multiple samples from individuals? I found these plots difficult to interpret; I think focusing on change from baseline would be much clearer (eg, replacing 1E with a plot of changes from baseline for active and control). Fig 1F seems to suggest there are a small number of placebo individuals who are different to everyone else— can you comment?

- **RESPONSE:** Word “clearly” definitely fit better. Line 267 (now line 317) changed to: “The PGCp group was clearly discernible, ... ”. The results presented in Figure 1 consider the paired nature of the before and after treatment samples in the PGCp vs GFDp and PGCd vs GFDd comparisons. Figure 1 was changed now to reflect this. We acknowledge that Figure 1E is non-informative and now replaced with plot that shows Correlation profile of all detected genes (n = 10063) log₂FC between PGCp VS GFDp and PGCp VS PGCd comparisons. In Figure 1F, we observe that patients in the placebo group with smaller VH:CrD after gluten challenge appear to be more distinct from the other groups than patients with higher VH:CrD. We hypothesize that the transcriptomic changes reflect the level of intestinal damage.

- Table 3: The statement “In the placebo group, the statistically significant decline in VH:CrD was similar across all the genotype groups” doesn’t look obviously true from this table Also you can’t infer a difference in tmt effect (ie HLA-tmt interaction) from whether individual differences are significant. You could formally test this by fitting appropriate models, which I assume is what your ANCOVA analysis is doing but I can’t really tell: write out the corresponding regression models and be clear about what null hypotheses is being tested for each p-value. Is any allowance for multiplicity made/needed? (1762 suggest Bonferroni correction, not clear how many tests). Also do some checks to show distributional assumptions are satisfied. Finally, can you be clearer about what is tabulated ; eg are values means +/- sd or something else? I assume GFD is the baseline value for the specified group but please confirm.

- **RESPONSE:** We completely agree with this comment. Repeated measures ANOVA was employed to assess the impact of treatment on VH:CrD within different timepoints (GFD and PGC) across HLA-DQ genetic background groups (G1, G2, and

G3). Table 3 now additionally contains interaction terms obtained from performed analysis.

Corresponding regression models for both Repeated measures ANOVA and ANCOVA models, accompanied with corresponding null hypothesis are added to Statistical analysis subsection of MATERIAL AND METHODS, lines 229-264.

Concerning multiplicity, line 262-264 now says: "To address multiple testing, the Bonferroni correction was applied to P-values (total tests performed = 3).

Distributional assumptions and detailed calculations for each model fitted are now included to separate supplemental file "Supplemental Materials and Methods.html."

Caption of figure 4 is modified to describe tabulated values: for fig.4A it is mean \pm SD; fig.4B and fig 4D have estimated marginal means \pm 95% CI plotted; fig.4C has VH:CrD ratios plotted together with estimate \pm 95% CI on top panel; fig.4E demonstrated expression in count, presented as mean + SD.

GFD is the baseline value for the specified group, we modified fig. 4 labels and captions, so now GFDd refers to baseline value for drug group and GFDp to baseline value for placebo group.

- L500 "ZED1227 does not seem to have any prominent direct adverse side effects" This seems a very strong statement. How do you conclude this from a transcriptomic analysis?

- **RESPONSE:** We acknowledge that the wording used in that sentence was strong. Based on the available data, we can state that "ZED1227 does not appear to induce significant transcriptomic changes in the organoid model." Line 500 (now line 576) is modified.

Decision Letter, first revision:

5th Dec 2023

Dear Dr. Viiri,

We have now finished reviewing your manuscript and response to reviewers comments on your manuscript entitled "Transglutaminase 2 inhibitor protects from gluten-induced intestinal damage in celiac disease – Transcriptomic analysis of a randomized gluten challenge study", reference number NI-A36062B.

Although the editors thought that the manuscript was interesting enough to send out for in-depth review, we remain concerned that the comments of Reviewer #2 would not be addressed in your revision plan and therefore we cannot accept the manuscript for publication. We would be willing to reconsider a revised version of the manuscript containing additional analysis of transcriptomic data and a replication cohort, if the data are available.

Although we cannot offer to publish your manuscript, we have discussed your manuscript and the referee feedback with our colleague Ildiko Gyory at Nature Communications. She expressed interest in your work and is happy for you to transfer your manuscript to Nature Communications. Ildiko tells us that she is comfortable about the significance and advance of the findings as long as the caveats of the study design are appropriately addressed. It is important that the ethical aspects of the challenge study are covered, but inclusion of another cohort is not required. Ildiko wishes to have a predictive point-by-point response, at the time of manuscript transfer, detailing the empirical and textual revisions that you will be providing to address the reviewers' points. Ildiko will then analyse the predictive response, and discuss with you, if necessary, before the next editorial decision.

We realize that this is disappointing. I hope that you continue to consider Nature Immunology for your results most significant for the immunology community and wish you well in your future investigations.

Sincerely,

Stephanie Houston, PhD
Senior Editor
Nature Immunology

Reviewers' comments:

Reviewer #1 (Remarks to the Author):

The authors have overall well addressed the issues raised and the manuscript despite some limitations merits publication in nature immunology. I have one comment related to the section result of the abstract that is difficult to follow as it is currently written. I would suggest to provide the number of

DEG between GFD (glute free diet) and placebo treated group and then to state that there is only one DEG when patients receive the TG2 inhibitor. The writing may benefit from some additional polishing,

Reviewer #2 (Remarks to the Author):

This paper followups their important NEJM clinical trial demonstrating histomorphometric benefit of their tissue transglutaminase inhibitor, ZED1227, with gluten challenge. The primary new data is a bulk RNASeq analysis on their cohort, serially, upon gluten rechallenge. Specifically, they test the highest ZED1227 dose (PGCd, 100 mg/day, n = 34) compared with (PGCp, n=24). They also perform genotype analysis (clustering their cohort into 3 genotype classes, G1, G2 and G3) and enteroid analysis to examine the role of IFNg signaling.

The major advance of this study over their NEJM article pertains to the transcriptional findings, basically equating their 6 week gluten challenge with the maximal dose of ZED1227 (100 mg) to gold standard negatives (GFD). The demonstration of transcriptional changes in their PGCp group compared to the dietary gold standard and their PGCd (post-gluten challenge, drug) cohorts is an important finding which is unlikely to be replicated, given the ethics of deliberately providing celiac patients with 3 grams of gluten (PGCp, placebo). However, the limitations imposed by their trial with placebo arm limitations would not necessarily limit expanding the a) timelines (6 weeks may be too short a time to result in changes in CD4 memory) for their treatment arm analyses, b) drug dosage effects -- they only describe RNASeq results with the highest dose, and c) gluten dosage effect--the drug is given immediately before the gluten cookie (3 grams-with the average diet including ~12 grams of gluten). This is clearly not how the drug, if used eventually, would have potential therapeutic benefit, with gluten exposures occurring throughout the day. So in many ways, the absence of transcriptional changes represents an important, but likely only minimal bar for ZED1227 molecular value.

More broadly, a transcriptome-wide approach is likely not the most powerful means of testing for early changes. While pathway analyses are described, a more immunologic-aware pathway analyses that incorporates the multiple known pathways and cells that interact (IFNg producing T cells combined with intraepithelial lymphocytes inducing epithelial damage) should be performed.

The genetic inference is under-powered and should include a replication cohort.

Author Rebuttal, first revision:

See inserted PDF

Reviewer #1

The authors have overall well addressed the issues raised and the manuscript despite some limitations merits publication in nature immunology. I have one comment related to the section result of the abstract that is difficult to follow as it is currently written. I would suggest to provide the number of DEG between GFD (glute free diet) and placebo treated group and then to state that there is only one DEG when patients receive the TG2 inhibitor. The writing may benefit from some additional polishing,

- **Response:** We thank reviewer for her/his comments. We have rewritten the results section in the abstract as suggested by the reviewer.

Reviewer #2

This paper followups their important NEJM clinical trial demonstrating histomorphometric benefit of their tissue transglutaminase inhibitor, ZED1227, with gluten challenge. The primary new data is a bulk RNASeq analysis on their cohort, serially, upon gluten rechallenge. Specifically, they test the highest ZED1227 dose (PGCd, 100 mg/day, n = 34) compared with (PGCp, n=24). They also perform genotype analysis (clustering their cohort into 3 genotype classes, G1, G2 and G3) and enteroid analysis to examine the role of IFN γ signaling.

The major advance of this study over their NEJM article pertains to the transcriptional findings, basically equating their 6 week gluten challenge with the maximal dose of ZED1227 (100 mg) to gold standard negatives (GFD). The demonstration of transcriptional changes in their PGCp group compared to the dietary gold standard and their PGCd (post-gluten challenge, drug) cohorts is an important finding which is unlikely to be replicated, given the ethics of deliberately providing celiac patients with 3 grams of gluten (PGCp, placebo).

- **Response:** We thank reviewer for her/his comments. As a background info, with all due respect, our studies and results can in principle be replicated as these phase 2a proof-of-concept gluten challenge studies are well accepted by both patients and ethical committees. But, academically such a full clinical drug trial with transcriptomics as primary or secondary outcomes is practically impossible to fund. Our multicentric European NEJM 2021 proof-of-concept study, testing the effect of a transglutaminase 2 inhibitor, ZED1227, blocking specifically the gluten- and immunological CD4 T cell-driven disease outcomes, used the well-established gluten challenge design from previous proof-of-concept celiac disease drug trials (ChemoCentryx trial: Hamilton G, et al. *Gastroenterology* 2008;134:Suppl. 1:A-493 and Lahdeaho M-L, et al. *BMC Gastroenterology* 2011;11:129, Alvine trial: Lahdeaho M-L, et al. *Gastroenterology* 2014;146:1649-1658, and Celimmune trial: Lahdeaho M-L, et al. *Lancet Gastroenterol Hepatol* 2019;4:948-959), and also used more recently (ImmunogenX trial: Murray JA, et al. *Gastroenterology* 2022;163:1510-1521).

In these trials we do not need to challenge with a full daily portion of gluten that is normally ingested (10-25 g gluten daily) by patients before the diagnosis. In other words, we do not need to induce a full-blown disease, the so called flat duodenal mucosa, but only give that much gluten for that long that we know we will see a clinically significant positive gluten and CD4 T cell driven duodenal mucosal deterioration in the placebo arm, a morphological and immunological injury, measured using quantitative readouts: morphometry separately for morphological injury and inflammation. Now, in the NEJM paper we showed the effect of ZED1227 both on our hard data, mucosal injury, and clinical outcomes (symptoms, CD4 T cell driven serology, i.e. serum autoantibodies, extraintestinal manifestation i.e. liver injury, all

dependent on gluten challenge-activated CD4 +ve T cells, now with 3 g daily gluten for 6 weeks).

However, the limitations imposed by their trial with placebo arm limitations would not necessarily limit expanding the a) timelines (6 weeks may be too short a time to result in changes in CD4 memory) for their treatment arm analyses,

- **Response:** The placebo arm was well adequate as it clearly showed significant gluten-induced disease outcomes that can only be driven by activated CD4 T cells. Here we show the results of the 100 mg ZED1227 at the molecular level. Our pathology collaborator showed in this trial biopsies a significant increase in the density of CD3+ IELs, CD8+ IELs, gamma-delta IELs, Ki67+ IEL Tcell, CD4+ lamina propria cells, and lamina propria plasma cells, all blocked by 100 mg ZED1227 (manuscript in preparation).

b) drug dosage effects -- they only describe RNASeq results with the highest dose,

- **Response:** We agree, that including all tested doses could be beneficial for study. But, since the transcriptomic study was optional, RNA isolation was not performed for all drug groups. The rationale behind this decision was to focus on the drug group that exhibited the most significant improvement compared to the placebo group in order to investigate potential transcriptomic changes in that particular group. This approach allowed us to prioritize the analysis and utilize our resources effectively. That is mentioned in line 637, among study limitations.

and c) gluten dosage effect--the drug is given immediately before the gluten cookie (3 grams-with the average diet including ~12 grams of gluten). This is clearly not how the drug, if used eventually, would have potential therapeutic benefit, with gluten exposures occurring throughout the day. So in many ways, the absence of transcriptional changes represents an important, but likely only minimal bar for ZED1227 molecular value.

- **Response:** Please note, none of the many celiac disease drug pipelines of today is targeting the ultimate goal: replacing gluten-free diet and taking care of for example the mentioned 12 g of gluten per day. All candidate drugs are now tested to be used on top of gluten-free diet, zero gluten does not exist, inadvertent gluten ingestion is inevitable in normal life. Patients with mucosal injury and symptoms are recruited into so called real-world trials where celiacs are ingesting some 100 mg of gluten per day, some perhaps a gram. The ongoing ZED1227 phase 2b trial is one of them. Dosing with ZED1227 was not 'immediately' but 30 min before gluten intake. Investigation of different doses and dosing schedules in CeD patients following their GFD is subject of a currently ongoing clinical trial.

Moreover, we have shown that ZED1227-TG2 target engagement still exist 24h after dosing in the intestinal mucosa of these patients (Isola et al. "The Oral Transglutaminase 2 Inhibitor ZED1227 Accumulates in the Villous Enterocytes in Celiac Disease Patients during Gluten Challenge and Drug Treatment" Int J Mol Sci 2023).

More broadly, a transcriptome-wide approach is likely not the most powerful means of testing for early changes. While pathway analyses are described, a more immunologic-aware pathway analyses that incorporates the multiple known pathways and cells that interact (IFN γ producing T cells combined with intraepithelial lymphocytes inducing epithelial damage) should be performed.

- **Response:** We have now performed immunological pathway analyses. It is previously well reported that Peroxisome proliferator-activated receptor gamma (PPAR γ) is downregulated in celiac disease by gliadin. This leads to activation of NOS2 and elevation of nitric oxide levels and inflammation. We were excited to find that inhibiting the gliadin deamidation activity of TG2 by ZED1227, all these pathogenic immunological changes in CeD are prevented. Of note, these pathways are less corrected by ZED1227 in the high-risk CeD G1 genotype as seen now in modified Fig. 5E.

We have added following texts in the results and discussion section. Moreover, we have built one new figure panel displaying the results explained in the new chapter “ZED1227 prevents immunological pathways induced by gluten in CeD”

RESULTS

ZED1227 prevents immunological pathways induced by gluten in CeD

As gluten challenge caused significant IFN- γ response and concomitant upregulation of TG2 expression and activity we analyzed gluten challenge induced immunological pathway alterations and how ZED1227 can inhibit them. Peroxisome proliferator-activated receptor gamma (PPAR γ) has been shown to transrepress inflammatory responses^{48,49}. PPAR γ is downregulated in celiac mucosa⁵⁰ and this has been shown to be mediated by TG2 and gliadin⁵¹. We also found that PPAR γ gene expression (Fig. 4A) and corresponding signaling pathway (Fig. 4B) is significantly less active after gluten challenge in PGCP group when compared to GFD and PGCd groups. We also detected negative correlation between the expression of TG2 and PPAR γ and the expression of PPAR γ and IEL count (Fig. 4C). This suggests that mucosal inflammatory response, kept in check by PPAR γ , is lifted during the gluten challenge in CeD and this can be prevented with ZED1227.

PPAR γ inhibit the expression of proinflammatory cytokines and it also silences inducible nitric oxide (NO) synthase (iNOS/NOS2)⁵² and NOS2 is induced in active CeD patients mucosa mainly in macrophages and enterocytes^{53–55} leading to systemic increase of NO in the plasma⁵⁶.

NO is needed for the responsiveness of natural killer (NK) cell to the NK cell activating factor IL-12 which stimulates their cytotoxicity and IFN γ release⁵⁷. Our data show that ZED1227 can inhibit gluten challenge-induced NOS2 upregulation (Fig. 4D) resulting to overrepresentation of gene sets involved in NO-IL12 and NK cell-mediated cytotoxicity (Fig. 4E) pathways. Also pathways to antigen presentation and IgA production are normalized with ZED1227 (Fig. 4E). Analysis of immunological cell gene markers shows that ZED1227 inhibit the infiltration of the cell types (especially CD8+ T cells, Plasma cells, NK cells and macrophages) involved in aforementioned inflammatory responses.

DISCUSSION

Our data also corroborate the previous findings that gliadin together with active TG2 induces attenuated PPAR γ activity which together with concomitant increase of IFN γ lead to increased mucosal NO production and inflammation^{50,51,53–56}. We show here that by

inhibiting the gliadin deamidation activity of TG2, all these pathogenic immunological changes in CeD can be prevented (Fig. 4).

Figure 4. ZED1227 effect on immunological pathways. A) Expression of PPARG mRNA in the GFDd, GFDp, PGcd, and PGcp patient groups. P-values represent adjusted P-values for multiple testing with the Benjamini–Hochberg method (FDR). GFDd ($n = 34$), GFDp ($n = 24$), PGcd ($n = 34$), and PGcp ($n = 23$). B) Gene set Z-score analyses for the PPAR Signaling pathway from KEGG database gene set. For comparison of groups, mean GSZ score asymptotic P-values calculation was applied to our datasets. Statistical significance was defined as a $P < .05$. GFDd+p ($n = 58$), PGcd ($n = 34$), and PGcp ($n = 23$). C) Correlation plots for TG2 mRNA

expression (upper panel) and IEL density (number CD3+ cells per 100 enterocytes, lower panel) against PPRG mRNA expression. Pearson correlation coefficient is presented. D) Expression of NOS2 mRNA in the GFDd, GFDp, PGCd, and PGCp patient groups. P-values represent adjusted P-values for multiple testing with the Benjamini–Hochberg method (FDR). GFDd ($n = 34$), GFDp ($n = 24$), PGCd ($n = 34$), and PGCp ($n = 23$). E) Gene set Z-score analyses for selected KEGG, Biocarta and Reactome databases gene set. For comparison of groups, mean GSZ score asymptotic P-values calculation was applied to our datasets. Statistical significance was defined as a $P < .05$. GFDd+p ($n = 58$), PGCd ($n = 34$), and PGCp ($n = 23$). F) Heatmap for selected CeD-specific immune cells marker genes detected in Atlasy et al., 2022 study⁷⁷. List of marker genes used added as supplemental. Samples ordered by increasing IELs density, as depicted in the scatter charts above the heatmap. GFDd ($n = 34$), GFDp ($n = 24$), PGCd ($n = 34$), and PGCp ($n = 23$). Z-score of normalized expression is plotted. PCs – Plasma Cells, NK - Natural killer cells, Inf-MF - Inflammatory Macrophages.

The genetic inference is under-powered and should include a replication cohort

- **Response:** We agree and homozygotes for DQ2 is of special interest in future trial, this must be confirmed. Here we give for the first time an indication for personalized medicine, why some patients were “outliers” in term of ZED1227 effect.

Decision Letter, second revision:

15th Mar 2024

Dear Dr. Viiri,

Thank you for submitting your revised manuscript "Transglutaminase 2 inhibitor protects from gluten-induced intestinal damage in celiac disease – Transcriptomic analysis of a randomized gluten challenge study" (NI-A36062D). It has now been seen by the original referees and their comments are below. The reviewers find that the paper has been somewhat improved in revision, and therefore we'll be happy in principle to publish it in Nature Immunology, pending minor revisions to satisfy the referees' final requests and to comply with our editorial and formatting guidelines. Please tone down conclusions in line with reviewers comments.

We will now perform detailed checks on your paper and will send you a checklist detailing our editorial and formatting requirements in about a week. Please do not upload the final materials and make any revisions until you receive this additional information from us.

If you had not uploaded a Word file for the current version of the manuscript, we will need one before beginning the editing process; please email that to immunology@us.nature.com at your earliest convenience.

Thank you again for your interest in Nature Immunology Please do not hesitate to contact me if you have any questions.

Sincerely,

Stephanie Houston, PhD
Senior Editor
Nature Immunology

Reviewer #2 (Remarks to the Author):

The authors have been responsive in places, but unresponsive and off target in other areas:

Strengths:

- PPARG addition: This is a more compelling study adding PPARG, and it is very reassuring (albeit not completely novel) to see the decrease in PPARG in the PGCP group

Weaknesses:

- I found the comment in the reply letter regarding their pathology collaborators are examining various lymphocyte subset frequencies ('manuscript in preparation') off target for a Nature Immunology submission. Given the dominant MHC Class II associations in celiac disease, the major pathophysiologic questions in serial, non-exposure (no dietary gluten) to exposure (to gluten) transcriptomics analyses deals with the nature of memory CD4+ T cell (vs. IEL subsets) differentiating and proliferative effects.

- The sample size with these effect sizes is too low to justify Precision Medicine inference, especially in the absence of including or discussing diversity. The MHC region is the most diverse locus in the genome. While the reply letter acknowledges this, in the resubmission, continued reference to differences on genotype (lines 468-483) are made, with any qualifications made with respect to the very small numbers in many of the cells in Table 2. Given the numbers of comparisons made (allele dose, pre- and post- challenges), this inference is underpowered.
- (Minor-moderate) The authors are unresponsive to the points raised with respect to 'real-world' exposures to dietary gluten. The administration of 3 grams of a gluten cookie either 'immediate' (my review) vs. 30 minutes (author response) before ZED1227 administration ignores the underlying, larger point; namely, the reason why transglutaminase inhibitors are needed is inadvertent, unpredictable, throughout the day, ingestion of dietary gluten. The absence of transcriptional differences (ZED1227 vs. pGC) with the lower dose (3 gm vs. 10 gm of gluten) is a relative strength of their approach; the use of 10 gram exposures, I agree is not a good direction. However, longer term follow-up (resource restricted in reply letter) beyond 6 weeks I would have thought is feasible.
- (Minor) line 967: misspelling of PPARG

Author rebuttal, second revision:

Reviewer #2 (Remarks to the Author):

The authors have been responsive in places, but unresponsive and off target in other areas:

Strengths:

- PPARG addition: This is a more compelling study adding PPARG, and it is very reassuring (albeit not completely novel) to see the decrease in PPARG in the PGCP group

- **Response:** We thank reviewer for her/his comments and suggestions to extend immunological pathway analyses. Indeed, it was very reassuring to find both PPARG and its subordinate NOS2 both affected in PGCP group as these have been well documented to be affected in celiac disease in the literature. And the fact that ZED1227 was able to prevent these changes is strengthening the point that TG2-inhibition can efficiently inhibit inflammation induced by gluten.

Weaknesses:

- I found the comment in the reply letter regarding their pathology collaborators are examining various lymphocyte subset frequencies ('manuscript in preparation') off target for a Nature Immunology submission. Given the dominant MHC Class II associations in celiac disease, the major pathophysiologic

questions in serial, non-exposure (no dietary gluten) to exposure (to gluten) transcriptomics analyses deals with the nature of memory CD4+ T cell (vs. IEL subsets) differentiating and proliferative effects.

- **Response:** Thank you, we agree to the referee's comment. Our lymphocyte subset response is here off target and we meant that bulk RNA-sequencing is not the best way to study the nature of memory CD4+ T cells and IELs as immunohistochemistry studies would offer more targeted analyses.

- The sample size with these effect sizes is too low to justify Precision Medicine inference, especially in the absence of including or discussing diversity. The MHC region is the most diverse locus in the genome. While the reply letter acknowledges this, in the resubmission, continued reference to differences on genotype (lines 468-483) are made, with any qualifications made with respect to the very small numbers in many of the cells in Table 2. Given the numbers of comparisons made (allele dose, pre- and post- challenges), this inference is underpowered.

- **Response:** We have now also acknowledged the limitation of our study and raised the issue of small sample size in the *discussion* as well: "We recognize the limitations of this study. The patient cohort is relatively modest and characterized by an uneven distribution of HLA-DQ genotypes. This resulted in small G1 subgroups within both the drug and placebo cohorts, which may have implications for statistical power and the generalizability of our results and warrants further corroborative studies."

We modified the text in the *results*: "We were able to divide patients into three groups according to their DQ genotypes, with G1 being high and G3 being low gluten-response groups (Table 1). However, one should note that the group sizes are relatively small."

Also *abstract* has been modified to: "Our results, with the limited sample size, also suggest that CeD patients might benefit from an HLA-DQ2/8 stratification based on gene doses to maximally eliminate the IFN- γ -induced mucosal damage triggered by gluten."

- (Minor-moderate) The authors are unresponsive to the points raised with respect to 'real-world' exposures to dietary gluten. The administration of 3 grams of a gluten cookie either 'immediate' (my review) vs. 30 minutes (author response) before ZED1227 administration ignores the underlying, larger point; namely, the reason why transglutaminase inhibitors are needed is inadvertent, unpredictable, throughout the day, ingestion of dietary gluten. The absence of transcriptional differences (ZED1227 vs. pGC) with the lower dose (3 gm vs. 10 gm of gluten) is a relative strength of their approach; the use of 10 gram exposures, I agree is not a good direction. However, longer term follow-up (resource restricted in reply letter) beyond 6 weeks I would have thought is feasible.

- **Response:** The clinical drug trial published in N Engl J Med 2021 showed a proof-of-concept result of TG2 inhibitor working in attenuating gluten-induced ill health. Here we have studied the 100 mg ZED1227 efficacy on 3 g gluten challenge for 6 weeks at a transcriptomic level. Dr.

Falk Pharma ongoing trial will again address the real-world setting and we do not know whether there will be studies at a transcriptomic level in that study. And yes, most probably before having a drug at the market there will be more phase 2b and then phase 3 trials also with different prescriptions, times per day and different doses and with longer follow ups. Please notice that at this point even once a day of ZED-1227 seems to act for quite a long at the mucosal level. And we have shown that ZED1227-TG2 target engagement still exist 24h after dosing in the intestinal mucosa of these patients boding well for the long drug effect (Isola et al. "The Oral Transglutaminase 2 Inhibitor ZED1227 Accumulates in the Villous Enterocytes in Celiac Disease Patients during Gluten Challenge and Drug Treatment" Int J Mol Sci 2023).

- (Minor) line 967: misspelling of PPARG

- **Response:** corrected

Final Decision Letter:

Dear Dr. Viiri,

I am delighted to accept your manuscript entitled "Transcriptomic analysis of intestine following administration of a transglutaminase 2 inhibitor to prevent gluten-induced intestinal damage in celiac disease" for publication in an upcoming issue of Nature Immunology.

Over the next few weeks, your paper will be copyedited to ensure that it conforms to Nature Immunology style. Once your paper is typeset, you will receive an email with a link to choose the appropriate publishing options for your paper and our Author Services team will be in touch regarding any additional information that may be required.

Please note that *Nature Immunology* is a Transformative Journal (TJ). Authors may publish their

research with us through the traditional subscription access route or make their paper immediately open access through payment of an article-processing charge (APC). Authors will not be required to make a final decision about access to their article until it has been accepted. Find out more about Transformative Journals.

Authors may need to take specific actions to achieve compliance with funder and institutional open access mandates. If your research is supported by a funder that requires immediate open access (e.g. according to Plan S principles) then you should select the gold OA route, and we will direct you to the compliant route where possible. For authors selecting the subscription publication route, the journal's standard licensing terms will need to be accepted, including self-archiving policies. Those licensing terms will supersede any other terms that the author or any third party may assert apply to any version of the manuscript.

Your paper will be published online soon after we receive your corrections and will appear in print in the next available issue.

Also, if you have any spectacular or outstanding figures or graphics associated with your manuscript - though not necessarily included with your submission - we'd be delighted to consider them as candidates for our cover. Simply send an electronic version (accompanied by a hard copy) to us with a possible cover caption enclosed.

If you have not already done so, we strongly recommend that you upload the step-by-step protocols used in this manuscript to the Protocol Exchange. Protocol Exchange is an open online resource that allows researchers to share their detailed experimental know-how. All uploaded protocols are made freely available, assigned DOIs for ease of citation and fully searchable through nature.com. Protocols can be linked to any publications in which they are used and will be linked to from your article. You

can also establish a dedicated page to collect all your lab Protocols. By uploading your Protocols to Protocol Exchange, you are enabling researchers to more readily reproduce or adapt the methodology you use, as well as increasing the visibility of your protocols and papers. Upload your Protocols at www.nature.com/protocolexchange/. Further information can be found at www.nature.com/protocolexchange/about .

Please note that we encourage the authors to self-archive their manuscript (the accepted version before copy editing) in their institutional repository, and in their funders' archives, six months after publication. Nature Portfolio recognizes the efforts of funding bodies to increase access of the research they fund, and strongly encourages authors to participate in such efforts. For information about our editorial policy, including license agreement and author copyright, please visit www.nature.com/ni/about/ed_policies/index.html

Sincerely,

Stephanie Houston, PhD
Senior Editor
Nature Immunology